# A Central Limit Theorem for Differentially Private Query Answering

**Jinshuo Dong**
Department of Computer Science
Northwestern University and IDEAL*
jinshuo@northwestern.edu

**Weijie J. Su**
Department of Statistics and Data Science
University of Pennsylvania
suw@wharton.upenn.edu

**Linjun Zhang**
Department of Statistics
Rutgers University
linjun.zhang@rutgers.edu

## Abstract

Perhaps the single most important use case for differential privacy is to privately answer numerical queries, which is usually achieved by adding noise to the answer vector. The central question is, therefore, to understand which noise distribution optimizes the privacy-accuracy trade-off, especially when the dimension of the answer vector is high. Accordingly, an extensive literature has been dedicated to the question and the upper and lower bounds have been successfully matched up to constant factors [BUV18, SU17]. In this paper, we take a novel approach to address this important optimality question. We first demonstrate an intriguing central limit theorem phenomenon in the high-dimensional regime. More precisely, we prove that a mechanism is approximately *Gaussian Differentially Private* [DRS21] if the added noise satisfies certain conditions. In particular, densities proportional to $\mathrm{e}^{-\|x\|_p^\alpha}$, where $\|x\|_p$ is the standard $\ell_p$-norm, satisfies the conditions. Taking this perspective, we make use of the Cramer–Rao inequality and show a "uncertainty principle"-style result: the product of privacy parameter and the $\ell_2$-loss of the mechanism is lower bounded by the dimension. Furthermore, the Gaussian mechanism achieves the constant-sharp optimal privacy-accuracy trade-off among all such noises. Our findings are corroborated by numerical experiments.

## 1 Introduction

Introduced in [DMNS06], to date *differential privacy* (DP) is perhaps the most popular privacy definition. One of the most important applications of differential privacy is to answer numeric queries. Given a function $f$ of interest, which is also termed a query, our goal is to evaluate this (potentially vector-valued) query $f$ on the sensitive data. To preserve privacy, a DP mechanism $M$ working on a dataset $D$, in its simplest form, is defined as

$$M(D) = f(D) + tX. \tag{1}$$

Above, $X$ denotes the noise term and $t$ is a scalar, which together are selected depending on the properties of the query $f$ and the desired privacy level. Among these, perhaps the most popular examples are the Laplace mechanism and the Gaussian mechanism where the noise $X$ follows the Laplace distribution and the Gaussian distribution, respectively.

---

*the Institute for Data, Econometrics, Algorithms, and Learning

35th Conference on Neural Information Processing Systems (NeurIPS 2021).

Aside from privacy considerations, the most important criterion of an algorithm is arguably the estimation accuracy in the face of choosing, for example, between the Laplace mechanism or its Gaussian counterpart for a given problem. To be concrete, consider a real-valued query $f$ with sensitivity 1—that is, $\Delta f = \sup_{D,D'} |f(D) - f(D')| = 1$, where the supremum is over all neighboring datasets $D$ and $D'$. Assuming $(\varepsilon, 0)$-DP for the mechanism $M$, we are interested in minimizing its $\ell_2$ loss defined as

$$\mathrm{err}(M) := \mathbb{E}(M(D) - f(D))^2 = \mathbb{E}(tX)^2 = t^2 \mathbb{E}X^2.$$

This question is commonly[2] addressed by setting $X$ to a standard Laplace random variable and $t = \varepsilon^{-1}$ [DMNS06]. This gives $\mathrm{err}(M) = 2\varepsilon^{-2}$. Moving forward, we *relax* the privacy constraint from $(\varepsilon, 0)$-DP to $(\varepsilon, \delta)$-DP for some small $\delta$. The canonical way, which was born together with the notion of $(\varepsilon, \delta)$-DP, is to add Gaussian noise [DKM+06]. A well-known result demonstrates that Gaussian mechanism with $X$ being the standard normal and $t = \frac{1}{\varepsilon}\sqrt{2\log(1.25\delta^{-1})}$ is $(\varepsilon, \delta)$-DP (see, e.g., [DR14]). The $\ell_2$-loss is $\mathrm{err}(M) = t^2 = 2\varepsilon^{-2} \cdot \log(1.25\delta^{-1})$.

A quick comparison between the two errors reveals a surprising message. The latter error $2\varepsilon^{-2} \cdot \log(1.25\delta^{-1})$ is larger than the former $2\varepsilon^{-2}$. In fact, the extra factor $\log(1.25\delta^{-1})$ is already greater than 10 when $\delta = 10^{-5}$. At least on the surface, this observation contradicts the fact that $(\varepsilon, \delta)$-DP is a relaxation of $(\varepsilon, 0)$-DP. Put differently, moving from Laplace to Gaussian, both privacy and accuracy get worse. Nevertheless, this contradiction suggests that we need a better alternative to the Gaussian mechanism instead of giving up the notion of $(\varepsilon, \delta)$-DP. Indeed, the truncated Laplace mechanism has been proposed as a better alternative to achieve $(\varepsilon, \delta)$-DP [GDGK20], which outperforms the Laplace mechanism in terms of estimation accuracy [3].

Motivated by these facts concerning the Laplace, Gaussian, and truncated Laplace mechanisms, one cannot help asking:

(Q1) Why was the truncated Laplace mechanism not considered in the first place? Are there any insights behind the design of such mechanisms?

(Q2) More importantly, are these insights inherent for answering one-dimensional queries, or can we extend them to high-dimensional setting?

In this paper, we tackle these fundamental questions, beginning with explaining (Q1) in Section 2 from the decision-theoretic perspective of DP [WZ10, KOV17, DRS21]. However, our main focus is (Q2). In addressing this question, we uncover a seemingly surprising phenomenon — it is impossible to utilize the $(\varepsilon, \delta)$ privacy budget in high-dimensional problems the same way as the truncated Laplace mechanism utilizes it in the one-dimensional problem. More specifically, we show a central limit behavior of the noise-addition mechanism in high dimensions, which, roughly speaking, says that for general noise distributions, the corresponding mechanisms *all* behave like a Gaussian mechanism. The formal language of "a mechanism behaves like the Gaussian mechanism" has been set up in [DRS21], where a notion called *Gaussian Differential Privacy* (GDP) was proposed. Roughly speaking, a mechanism is $\mu$-GDP if it offers as much privacy as adding $N(0, \mu^{-2})$ noise to a sensitivity-1 query. As in the $(\varepsilon, \delta)$-DP case, the smaller $\mu$ is, the stronger privacy is offered.

To state our first main contribution, let $f$ be an $n$-dimensional query and assume that its $\ell_2$-sensitivity is 1. Consider the noise addition mechanism $M(D) = f(D) + tX$ where $X$ has a log-concave density $\propto e^{-\varphi(x)}$ on $\mathbb{R}^n$. Let $\mathcal{I}_X := \mathbb{E}[\nabla\varphi(X)\nabla\varphi(X)^T]$ be the $n \times n$ Fisher information matrix and $\|\mathcal{I}_X\|_2$ be its operator norm.

**Theorem 1.1** (Central Limit Theorem (Informal version of Theorem 3.1)). *Under certain conditions on $\varphi$, for $t = \mu^{-1} \cdot \sqrt{\|\mathcal{I}_X\|_2}$, the corresponding noise addition mechanism $M$ defined in Eq.(1) is asymptotically $\mu$-GDP as the dimension $n \to \infty$ except for an $o(1)$ fraction of directions of $f(D) - f(D')$.*

In particular, the norm power functions $\varphi(x) = \|x\|_p^\alpha$ $(p, \alpha \geqslant 1)$ satisfy these technical conditions. Note that this class already contains correlated noise, so the results in [DRS21] do not apply here. Numerical results in Figure 1 shows that the convergence occurs for a dimension as small as 30.

---

[2] If $f$ is integer-valued, then the doubly geometric distribution is a better choice and yields an $\ell_2$-loss of $\frac{1}{2}\sinh^{-2}\frac{\varepsilon}{2} < 2\varepsilon^{-2}$. In the so-called high privacy regime, i.e. $\varepsilon \to 0$, the two $\ell_2$-losses have the same order in the sense that their ratio goes to 1.

[3] One may blame the sub-optimality of the choice of $t$, but the problem remains even if the smallest possible $t$ from [BW18] is applied.

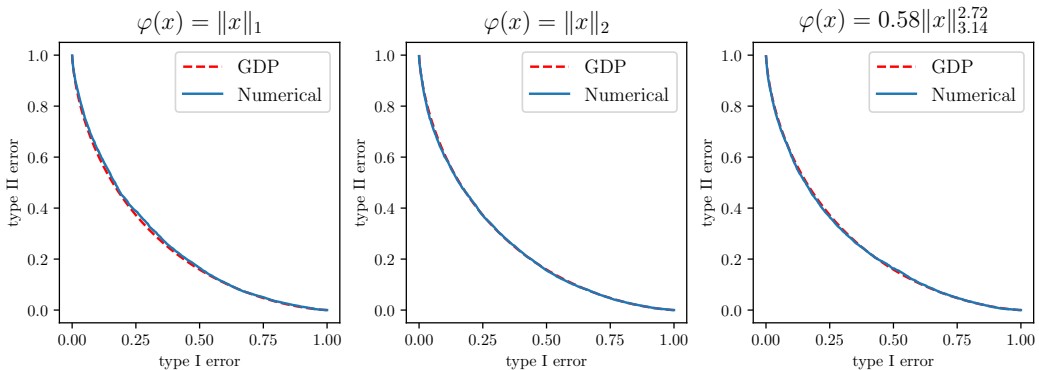

Figure 1: Fast convergence to GDP as claimed in Theorem 1.1. Blue solid curves indicate the true privacy (i.e. ROC functions, see Section 2 for details) of the noise addition mechanism considered in Theorem 1.1. Red dashed curves are GDP limit predicted by our CLT. In all three panels the dimension $n = 30$. Numerical details can be found in the appendix.

We then elaborate on the condition "$o(1)$ fraction of $f(D) - f(D')$". Following the original definition, DP or GDP is a condition that needs to hold for arbitrary neighboring datasets $D$ and $D'$. This worst case perspective is exactly what prevents us to observe the central limit behavior. For example, consider a certain pair of datasets with $f(D) = (0, 0, \ldots, 0)$ and $f(D') = (1, 0, \ldots, 0)$, then privacy is completely determined by the first marginal distribution of $X$, and the dimension $n$ plays no role here. The "$o(1)$ fraction of $f(D) - f(D')$" rules out the essentially low-dimension cases and reveals the truly high-dimensional behavior.

In summary, Theorem 1.1 suggests that when the dimension is high, a large class of noise addition mechanisms behave like the Gaussian mechanism, and hence are doomed to a poor use of the given $(\varepsilon, \delta)$ privacy budget, in the same fashion as we have seen in the one-dimensional example.

However, admitting the central limit phenomenon, our second theorem turns the table and characterizes the optimal privacy-accuracy trade-off and justifies the Gaussian mechanism. To see this, recall that the noise addition mechanism defined in Equation (1) is determined by the pair $(t, X)$. Both privacy and accuracy are jointly determined by $t$ and $X$. Adopting the central limit theorem 1.1, it is convenient to take an equivalent parametrization, which is $(\mu, X)$, where $\mu$ is the desired (asymptotic) GDP parameter. Given $X$, the two parametrizations are related by $t = \mu^{-1} \cdot \sqrt{\|\mathcal{I}_X\|_2}$. Using parameters $(\mu, X)$, the corresponding mechanism $M_{\mu, X}$ is given by

$$M_{\mu, X}(D) = f(D) + \mu^{-1} \cdot \sqrt{\|\mathcal{I}_X\|_2} \cdot X$$

By Theorem 1.1, it is asymptotically $\mu$-GDP. The following theorem states in an "uncertainty principle" fashion that the privacy parameter and the error cannot be small at the same time.

**Theorem 1.2.** *As long as the Fisher information of $X$ is defined, we have*

$$\mu^2 \cdot \mathrm{err}(M_{\mu, X}) \geqslant n.$$

*The equality holds if $X$ is $n$-dimensional standard Gaussian.*

Combining Theorems 1.1 and 1.2, among all the noise that satisfies the conditions of Theorem 1.1, Gaussian yields the constant-sharp optimal privacy-accuracy trade-off. As far as we know, this is the first result characterizing optimality with the sharp constant when the dimension is high.

The privacy conclusion of Theorem 1.1 does not work for every pair of neighboring datasets, so it is worth noting that we do NOT intend to suggest this as a valid privacy guarantee. Instead, we present it as an interesting phenomenon that has been largely overlooked in the literature. Furthermore, this central limit theorem admits an elegant characterization of privacy-accuracy trade-off that is sharp in constant. From a theoretical point of view, the proof of Theorem 1.1, as we shall see in later sections, involves *non-linear* functionals of high dimensional distributions. This type of results are, to the best of our knowledge, quite underexplored compared to linear functionals, so our results may serve as an additional motivation to study this type of questions.

**Related work**   There is a large body of literature on the characterization of privacy-accuracy trade-off for query answering mechanisms. For the one-dimensional case, the constant-sharp optimal noise for $(\varepsilon, 0)$-DP was shown to have a piece-wise constant density by [GV16]. This complements our discussion in Figure 2. When the dimension is high, only up-to-constant-factor optimality was known. In particular, [BUV18, SU17] confirm that Gaussian mechanism is minimax rate optimal under $(\varepsilon, \delta)$-DP by a novel lower bound technique. In addition, [ENU20] also confirms the minimax optimality of Gaussian mechanism for linear queries with a refined notion of sensitivity. Our work extends this direction by taking the CLT perspective and providing an elegant constant-sharp optimality result. There are also works studying the up-to-constant-factor minimax optimality in other models, such as the (sparse) linear regression [CWZ21], generalized linear models [CWZ20, SSTT21], Gaussian mixtures [KSSU20, ZZ21] and so on. In our work, we initialize the investigation in the (simpler) mean estimation problem, and leave the constant-sharp optimality in other problems for future work.

## 2   GDP and the ROC Functions

The decision theoretic interpretation of DP was first proposed in [WZ10] and then extended by [KOV17]. More recently, [DRS21] systematically studied this perspective and developed various tools. In this section we take this perspective and introduce the basics of [DRS21]. This will allow us to give an intuitive answer to (Q1).

Suppose each individual's sensitive information is an element in the abstract set $\mathcal{X}$. A dataset $D$ of $k$ people is then an element in $\mathcal{X}^k$. Let a randomized algorithm $M$ take a dataset as input and let $D$ and $D'$ be two neighboring datasets, i.e. they differ by one individual. Differential privacy seeks to limit the power of an adversary identifying the presence of an arbitrary individual in the dataset. That is, with the output as the observation, telling apart $D$ and $D'$ must be hard for the adversary. Decision theoretically, the quality of an attack is measured by the errors it makes. The more error it is forced to make, the more privacy $M$ provides.

To breach the privacy, the adversary performs the following hypothesis testing attack:

$$H_0 : \text{output} \sim M(D) \quad \text{vs} \quad H_1 : \text{output} \sim M(D').$$

By the random nature of $M$, $M(D)$ and $M(D')$ are two distributions. We emphasize this point by denoting them by $P$ and $Q$. The errors mentioned above are simply the probabilities confusing $D$ and $D'$, which are commonly known as false positive and false negative rates. Because of the symmetry of the neighboring relation, there is no need to worry about which is which.

**ROC function.**   For simplicity assume $M$ outputs a vector in $\mathbb{R}^n$. A general decision rule for testing $H_0$ against $H_1$ has the form $\phi : \mathbb{R}^n \to \{0, 1\}$. Observing $v \in \mathbb{R}^n$, hypothesis $H_i$ is accepted if $\phi(v) = i$, for $i \in \{0, 1\}$. The false positive rate (type I error) of $\phi$, i.e. mistakenly accepting $H_1 : v \sim M(D') = Q$ while actually $v \sim M(D) = P$, is $\alpha_\phi := \mathbb{P}_{v \sim P}(\phi(v) = 1) = \mathbb{E}_P(\phi)$. Similarly, the false negative rate (type II error) of $\phi$ is $\beta_\phi := 1 - \mathbb{E}_Q(\phi)$. Note that both errors are in $[0, 1]$. Consider the function $f_{P,Q} : [0, 1] \to [0, 1]$ defined as follows:

$$f_{P,Q}(\alpha) := \inf\{1 - \mathbb{E}_Q(\phi) : \phi \text{ satisfies } \mathbb{E}_P(\phi) \leqslant \alpha\}. \tag{2}$$

That is, $f_{P,Q}(\alpha)$ equals the minimum false negative rate that one can achieve when false negative is at most $\alpha$. The graph of $f_{P,Q}$ is exactly the flipped ROC curve of the family of optimal tests (which, by Neyman–Pearson lemma, are the likelihood ratio tests). We call it the *ROC function of the test $P$ vs $Q$*. The same notion is called *trade-off function of $P$ and $Q$* in [DRS21] and is denoted by $T[P, Q]$. We avoid this name because in our paper "trade-off" mainly refers to the privacy-accuracy trade-off, but we will keep their notation.

**DP and ROC function**   Plugging in the privacy context where $P = M(D), Q = M(D')$, from the discussion above, we see that $T[M(D), M(D')]$ measures the optimal error distinguishing $M(D)$ and $M(D')$. Therefore, a lower bound on $T[M(D), M(D')]$ implies privacy of $M$. Indeed, [WZ10, KOV17] showed that $M$ is $(\varepsilon, \delta)$-DP if and only if $T[M(D), M(D')] \geqslant f_{\varepsilon, \delta}$ pointwise in $[0, 1]$ for any neighboring dataset $D, D'$. The graph of $f_{\varepsilon, \delta}$ is plotted in the left panel of Figure 2. Compared to a single $(\varepsilon, \delta)$ bound, the ROC function $T[M(D), M(D')]$ provides a more refined picture of the privacy of $M$. In fact, [DRS21] shows that the ROC function is equivalent to an infinite family of $(\varepsilon, \delta)$ bounds, which is called privacy profile in [BBG20].

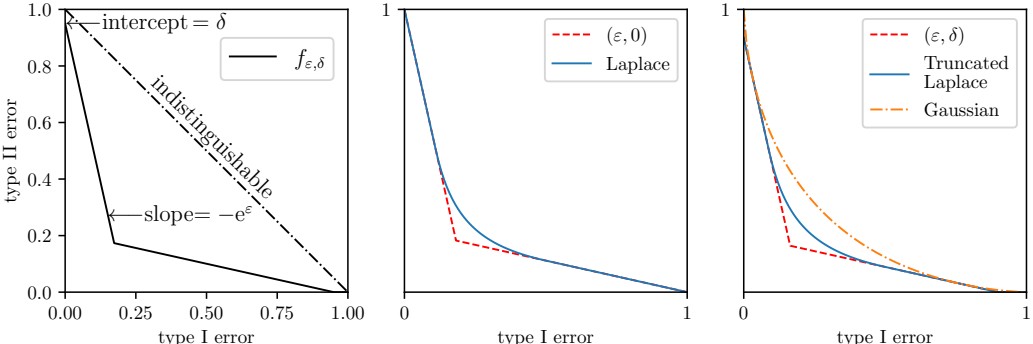

Figure 2: Left: $f_{\varepsilon,\delta}$ which recovers the classical $(\varepsilon,\delta)$-DP definition. Middle: Laplace mechanism is $(\varepsilon,0)$-DP Right: Gaussian mechanism and truncated Laplace mechanism are both $(\varepsilon,\delta)$-DP

**Truncated Laplace vs Gaussian through the lense of ROC function**  Now we use ROC function to answer (Q1) in the introduction. Namely, we want to explain the embarrassing situation of the Gaussian mechanism that the privacy budget is not fully used, and the success of the truncated Laplace mechanism.

When $M$ is the Laplace mechanism which is designed to be $(\varepsilon,0)$-DP, it is not hard to determine $T[M(D),M(D')]$ via Neyman–Pearson lemma and verify that it is indeed lower bounded by $f_{\varepsilon,0}$ (see the middle panel of Figure 2). In fact, $T[M(D),M(D')]$ mostly agrees with $f_{\varepsilon,0}$. In other words, the $(\varepsilon,0)$ privacy budget is almost[4] fully utilized.

When $M$ is the Gaussian mechanism with $(\varepsilon,\delta)$-DP gaurantee, $T[M(D),M(D')]$ is naturally lower bounded by $f_{\varepsilon,\delta}$, however, there is a large gap between the two curves (see the right panel of Figure 2). The $(\varepsilon,\delta)$ privacy budget is poorly utilized by the Gaussian mechanism. This explains why the $l^2$-loss of Gaussian mechanism is not satisfactory.

For noise addition mechanism, if the noise is bounded, say a uniform $[-1,1]$ distribution, then $T[M(D),M(D')] = f_{0,\delta}$ for some $\delta \in (0,1)$. This suggests us to consider bounded noise if we want to add a $\delta$ slack in privacy to the Laplace mechanism. The obvious attempt is then to truncate Laplace noise. Indeed, the corresponding ROC function is as close to $f_{\varepsilon,\delta}$ as that of the Laplace mechanism to $f_{\varepsilon,0}$ (also see the right panel of Figure 2). This not only explains the success of the truncated Laplace mechanism, but also points us to the right direction in searching for such a mechanism.

In hindsight, this achievement for one-dimensional mechanisms is due to the following fact: as we change the noise distribution, the corresponding ROC functions are significantly different. Hence we can pick the one that best utilizes our privacy budget. However, in the next section we will argue that this no longer works when the dimension is high — many (if not all) choices of noise distribution yield the same ROC function, which is the ROC of Gaussian mechanism.

**ROC function of the Gaussian mechanism**  For $\mu \geqslant 0$, let $G_\mu := T[\mathcal{N}(0,1), \mathcal{N}(\mu,1)]$ where $\Phi$ denotes the cumulative distribution function (CDF) of the standard normal distribution. Consider a query $f$ with sensitivity 1 and let $\mathrm{Lap}(0,1)$ be the standard Laplace noise. Just like $\varepsilon$-DP captures the privacy of the mechanism $M(D) = f(D) + \varepsilon^{-1} \cdot \mathrm{Lap}(0,1)$, the function $G_\mu$ captures the privacy of $M(D) = f(D) + \mu^{-1} \cdot N(0,1)$. In fact, if $f(D') - f(D) = 1$, then $M(D) = N(f(D), \mu^{-2})$ and $M(D') = N(f(D'), \mu^{-2})$. By its hypothesis testing construction, $T[P,Q]$ remains invariant when an invertible transformation is simultaneously applied to $P$ and $Q$, resulting in

$$T[M(D), M(D')] = T[N(f(D), \mu^{-2}), N(f(D'), \mu^{-2})] = T[N(0,1), N(\mu,1)] = G_\mu$$

Therefore, the privacy of a Gaussian mechanism is precisely captured by the ROC function $G_\mu$. A general mechanism $M$ is said to be *Gaussian differentially private* (GDP) if it offers more privacy than a Gaussian mechanism. More specifically,

**Definition 2.1** (GDP). *An algorithm $M$ is $\mu$-GDP if $T[M(D), M(D')] \geqslant G_\mu$ for any pair of neighboring datasets $D$ and $D'$.*

---

[4]If the query is integer-valued, then $(\varepsilon,0)$ privacy budget can be saturated by adding doubly geometric noise.

Alternatively, $M$ is $\mu$-GDP if and only if $\inf_{D,D'} T[M(D), M(D')] \geqslant G_\mu$ where the infimum of ROC functions is interpreted pointwise, and the infimum is taken over all neighboring datasets $D$ and $D'$. This inequality says $M$ offers more privacy than the corresponding Gaussian mechanism. If the equality holds, i.e.

$$\inf_{D,D'} T[M(D), M(D')] = G_\mu \tag{3}$$

then it means the mechanism $M$ offers exact the same amount of privacy as the corresponding Gaussian mechanism. In fact, the CLT to be presented in the next section has this flavor of conclusion.

## 3 Central Limit Theorem

In the following two sections we turn to addressing (Q2). This section is dedicated to the rigorous form of the CLT and the discussion.

The experience with the CLT for i.i.d. random variables suggests that the statement for the normalized special case is usually the most comprehensible. Therefore, we will state the normalized version as Theorem 3.1 and derive the general case as Corollary 3.2, which is also the rigorous version of our informal theorem 1.1 mentioned in the introduction.

Consider an $n$-dimensional query $f : \mathcal{X}^k \to \mathbb{R}^n$. We assume it has $\ell_2$-sensitivity 1, i.e. $\sup_{D,D'} \|f(D) - f(D')\|_2 = 1$. Suppose $\varphi : \mathbb{R}^n \to \mathbb{R}$ is convex and $\mathrm{e}^{-\varphi}$ is integrable on $\mathbb{R}^n$. The log-concave random vector with density $\propto \mathrm{e}^{-\varphi(x)}$ will be denoted by $X_\varphi$. Define the function class

$$\mathfrak{F}_n := \{\varphi : \mathbb{R}^n \to \mathbb{R} \text{ convex} \mid \varphi(x) = \varphi(-x), \mathrm{e}^{-\varphi} \in L^1(\mathbb{R}^n), \mathbb{E}\|X_\varphi\|_2^2 < +\infty, \mathbb{E}\|\nabla\varphi(X_\varphi)\|_2^2 < +\infty\}.$$

The regularity conditions guarantee that $X_\varphi$ has finite second moments and Fisher information matrix defined as $\mathcal{I}_\varphi = \mathbb{E}[\nabla\varphi(X_\varphi)\nabla\varphi(X_\varphi)^T]$. Furthermore, we also have $\mathbb{E}X_\varphi = 0$ by symmetry and $\mathbb{E}\nabla\varphi(X_\varphi) = 0$ by standard theory of Fisher information (e.g. [VdV00]). We will focus on this class of functions for the rest of paper.

The $n$-dimensional noise addition mechanism of interest takes the form $M(D) = f(D) + tX_\varphi$. The parameter $t$ is only for the convenience of tuning and can be absorbed into $\varphi$. In fact, $tX_\varphi$ has log-concave density $\propto \mathrm{e}^{-\varphi(x/t)}$, so it is distributed as $X_{\tilde{\varphi}}$ where $\tilde{\varphi}(x) = \varphi(x/t)$. For the normalized CLT, we set $t = 1$ and assume $\mathcal{I}_\varphi$ is the $n \times n$ identity matrix $I_{n \times n}$.

Since what we are going to present is an asymptotic result where the dimension $n \to \infty$, the above objects necessarily appear with an index $n$, i.e. we have $f_n, \varphi_n, X_{\varphi_n}$ and $\mathcal{I}_{\varphi_n}$. The latter two are often denoted by $X_n$ and $\mathcal{I}_n$ for brevity. With normalization, the $n$-dimensional mechanism of interest is $M_n(D) = f_n(D) + X_n$. For clarity, we choose to state the theorem first, and then present the details of the technical conditions.

**Theorem 3.1.** *If the function sequence $\varphi_n$ satisfies conditions (D1) and (D2), then there is a sequence of positive numbers $c_n$ with $c_n \to 0$ as $n \to \infty$ and a subset $E_n \subseteq S^{n-1}$ with $\mathbb{P}_{v \sim S^{n-1}}(v \in E_n) > 1 - c_n$ such that*

$$\| \inf_{D,D'} T[M_n(D), M_n(D')] - G_1 \|_\infty \leqslant c_n$$

*where the infimum is taken over $D, D'$ such that $\frac{f_n(D') - f_n(D)}{\|f_n(D') - f_n(D)\|_2} \in E_n$.*

Here $v \sim S^{n-1}$ means $v$ comes from a uniform distribution of the unit sphere $S^{n-1} \subseteq \mathbb{R}^n$. The conclusion is basically that $\inf_{D,D'} T[M_n(D), M_n(D')] \to G_1$, i.e. $M_n$ is asymptotically GDP. Similar to the interpretation of (3), it means the mechanism $M_n$ provides the same amount of privacy as a Gaussian mechanism in the limit of $n \to \infty$. However, a fraction of neighboring datasets has to be excluded. More specifically, the limit holds if the direction of the difference $f_n(D') - f_n(D)$ falls in $E_n$, an "almost sure" event as the dimension $n \to \infty$. As we remarked in the introduction, directions in $E_n$ can exhibit low dimensional behavior and hence must be ruled out for any high-dimensional observation.

For a vector $v \in \mathbb{R}^n$ and $\varphi \in \mathfrak{F}_n$, let $P_v^\varphi : \mathbb{R}^n \to \mathbb{R}$ be defined as $P_v^\varphi(x) = \varphi(x + v) - \varphi(x) - \frac{1}{2}v^T\mathcal{I}_\varphi v$. For two random variables $X$ and $Y$, their Kolmogorov–Smirnov distance $\mathrm{KS}(X, Y)$ is defined as the $\ell_\infty$ distance of their CDFs. A sequence of random variables is denoted by $o_P(1)$ if they converge in probability to 0. The technical conditions for the CLT are as follows. Note that each of them are conditions on the function sequence $\varphi_n$.

(D1) $\mathrm{KS}\big(P_v^{\varphi_n}(X_n), v^T \nabla\varphi_n(X_n)\big) = o(1)$ with probability at least $1 - o(1)$ over $v \sim S^{n-1}$

(D2) $\|\nabla\varphi_n(X_n)\|_2 = \sqrt{n} \cdot (1 + o_P(1))$

**Remark 1.** Dropping the cumbersome subscripts $n$, (D1) roughly asks that

$$P_v^\varphi(X) = \varphi(X + v) - \varphi(X) - \tfrac{1}{2}v^T \mathcal{I}_\varphi v \approx v^T \nabla\varphi(X)$$

Since $\mathcal{I}_\varphi$ is the expectation of the Hessian of $\varphi$, we see that (D1) is basically a regularity condition stating that the Taylor expansion of $\varphi$ holds on average up to the second order.

**Remark 2.** Condition (D2) basically says that $\nabla\varphi(X)$ mostly falls on a spherical shell of radius $\sqrt{n}$ (as it should since $\mathcal{I}_\varphi = \mathbb{E}[\nabla\varphi(X)\nabla\varphi(X)^T]$ is assumed to be identity). A deeper understanding is provided by an alternative interpretation of condition (D1), using a new notion we propose called "likelihood projection".

**Likelihood Projection.** The function $P_v^\varphi$ defined above is called the "likelihood projection" along direction $v$. It is (up to an additive constant) the log likelihood ratio of $X_\varphi$ and its translation $X_\varphi - v$. In fact, $X_\varphi$ has density $\frac{1}{Z_\varphi}\mathrm{e}^{-\varphi(x)}$ and $X_\varphi - v$ has density $\frac{1}{Z_\varphi}\mathrm{e}^{-\varphi(x+v)}$ where $Z_\varphi$ is the common normalizing constant. The log likelihood ratio is $\varphi(x+v) - \varphi(x)$. This explains the word "likelihood". To observe its nature as a "projection", consider the special case $\varphi(x) = \frac{1}{2}\|x\|_2^2$. Straightforward calculation suggests that $\mathcal{I}_\varphi$ is identity and $P_v^\varphi(x) = v^T x$. So it is indeed a generalization of the linear projection along direction $v$.

The alternative interpretation of condition (D1) is that when the dimension is high, the "likelihood projection" $P_v^\varphi(X)$ is roughly a linear projection to the direction $v$. Condition (D2) is then the "thin-shell" condition proposed in Sudakov's theorem [Sud78] which we state in the appendix as a necessary tool for the proof of our CLT.

For the general case, consider $M_n(D) = f_n(D) + t_n X_{\varphi_n}$ where $t_n = \mu^{-1} \cdot \sqrt{\|\mathcal{I}_n\|_2}$. The factor $\sqrt{\|\mathcal{I}_n\|_2}$ normalizes the Fisher information to the identity, and the factor $\mu^{-1}$ controls the final privacy level. For this mechanism, we have

**Corollary 3.2.** *If the function sequence $\tilde{\varphi}_n(x) = \varphi_n(\|\mathcal{I}_n\|_2^{-\frac{1}{2}}x)$ satisfies conditions (D1) and (D2) and that $\mathcal{I}_n = \|\mathcal{I}_n\|_2 \cdot (1 + o(1)) \cdot I_{n\times n}$, then there is a sequence of positive numbers $c_n \to 0$ and a subset $E_n \subseteq S^{n-1}$ for each $n$ with $\mathbb{P}_{v \sim S^{n-1}}(v \in E_n) > 1 - c_n$ such that*

$$\| \inf_{D,D'} T[M_n(D), M_n(D')] - G_\mu \|_\infty \leqslant c_n$$

*where the infimum is taken over $D, D'$ such that $\frac{f_n(D') - f_n(D)}{\|f_n(D') - f_n(D)\|_2} \in E_n$.*

In particular, when $p$ and $\alpha$ belong to $[1, +\infty)$, norm powers $\|x\|_p^\alpha$ satisfy the above conditions.

**Lemma 3.3.** *For $p \in [1, +\infty), \alpha \in [1, +\infty)$, let $c_{p,\alpha} = \alpha^{-1} \cdot p^{-\alpha + \frac{\alpha}{p}} \cdot \left(\frac{\Gamma(2 - \frac{1}{p})}{\Gamma(\frac{1}{p})}\right)^{-\frac{\alpha}{2}}$, the sequence of functions $\varphi_n(x) = n^{1 - \frac{\alpha}{p}} \cdot c_{p,\alpha}\|x\|_p^\alpha$ satisfies conditions (D1) and (D2) and that $\mathcal{I}_n = \|\mathcal{I}_n\|_2 \cdot (1 + o(1)) \cdot I_{n\times n}$.*

The parameter $c_{p,\alpha}$ and the power of $n$ are determined by the Fisher information, which can be found in Lemma 4.2. More generally, we conjecture that

**Conjecture 3.4.** *All functions in $\mathfrak{F}_n$ satisfy (D1) and (D2).*

Recall that $\varphi \in \mathfrak{F}_n$ lead to log-concave distributions. We limit the scope of our conjecture to log-concave distributions because of an interesting lemma involved in the proof of the central limit theorem 3.1. Consider the mechanism $M^t(D) = f(D) + tX$, with the emphasis on the scaling parameter $t$. As $t$ increases, $M^t$ obviously loses accuracy regardless of log-concavity of $X$. On the other hand, when it comes to privacy, we have

**Lemma 3.5.** *When $X$ has log-concave distribution and $t \geqslant 0$, the ROC function $T[M^t(D), M^t(D')]$ is (pointwise) monotone increasing in $t$ for any $D, D'$.*

Since larger ROC function means more privacy, this lemma confirms that $M^t$ gains privacy as $t$ increases. In other words, it confirms the existence of "privacy-accuracy trade-off" given the log-concavity of $X$. Note that without log-concavity, monotonicity in the lemma need not hold. For a

one-dimensional example, consider an $X$ that supports on even numbers and $f(D) = 0$, $f(D') = 2$. When $t = 2$, $T[M^t(D), M^t(D')] = T[2X, 2X + 2] = T[X, X + 1]$. There is no privacy in this case as $X$ and $X + 1$ has completely disjoint support. On the other hand, when $t = 1$, $T[M^t(D), M^t(D')] = T[X, X + 2]$ and incurs some privacy. That is, more noise does not imply more privacy, hence violating the conclusion of Lemma 3.5.

In summary, results in this section show that mechanisms adding noise that satisfies (D1) and (D2) (e.g. densities $\propto e^{-\|x\|_p^\alpha}$) behave like a Gaussian mechanism. Changing the noise in this class does not change the ROC function by much. Hence we cannot repeat the success at fully utilizing the $(\varepsilon, \delta)$ privacy budget as in Section 2.

On the other hand, our CLT involves Fisher information, and hence gives us the opportunity to relate to the (arguably) most successful tool for constant-sharp lower bound — the Cramer–Rao inequality. This will be the content of the next section.

## 4    Privacy-Accuracy Trade-off via Cramer–Rao Inequality

The central limit theorem in the previous section suggests that we use GDP parameter $\mu$ to measure privacy. Adopting this, we will show that the privacy-accuracy trade-off is naturally characterized by the Cramer–Rao lower bound. The conclusion has a similar flavor to the uncertainty principles.

Recall that the mechanism $M(D) = f(D) + tX_\varphi$ is determined by two "parameters": the shape parameter $\varphi \in \mathfrak{F}_n$ which determines the distribution of $X_\varphi$, and the scale parameter $t$. If $\varphi$ also satisfies the conditions of Theorem 3.1, then we can use the desired (asymptotic) GDP parameter $\mu$ to determine the scale parameter, i.e. $t = \mu^{-1} \cdot \sqrt{\|\mathcal{I}_\varphi\|_2}$. Using the equivalent parametrization $(\mu, \varphi)$, the corresponding mechanism $M_{\mu,\varphi}$ is given by

$$M_{\mu,\varphi}(D) = f(D) + \mu^{-1} \cdot \sqrt{\|\mathcal{I}_\varphi\|_2} \cdot X_\varphi. \tag{4}$$

As we have explained in the introduction, one way to measure the accuracy of the mechanism is the mean squared error of the noise

$$\operatorname{err}(M_{\mu,\varphi}) = \mathbb{E}\|tX_\varphi\|_2^2 = \mu^{-2} \cdot \|\mathcal{I}_\varphi\|_2 \cdot \mathbb{E}\|X_\varphi\|_2^2. \tag{5}$$

The following theorem characterizes the privacy-accuracy trade-off as the product of the mean squared error $\operatorname{err}(M_{\mu,\varphi})$ and privacy parameter $\mu^2$.

**Theorem 4.1** (Restating Theorem 1.2)**.** *For any $\varphi \in \mathfrak{F}_n$ and $M_{\mu,\varphi}$ defined as in* (4)*, we have*

$$\mu^2 \cdot \operatorname{err}(M_{\mu,\varphi}) \geqslant n.$$

*In addition, the equality holds if the added noise $X$ is $n$-dimensional standard Gaussian.*

*Proof of Theorem 4.1.* To simplify notations we will drop the subscript $\varphi$ in $X$. We first claim that it suffices to show the following uncertainty-principle-like result

$$\operatorname{Var}[X] \cdot \operatorname{Var}[\nabla\varphi(X)] \geqslant n^2. \tag{6}$$

where the notation $\operatorname{Var}[\cdot]$ is slightly abused to denote the mean squared distance of a random vector from its expectation, i.e. $\operatorname{Var}[X] = \mathbb{E}[\|X - \mathbb{E}X\|_2^2]$.

To see why (6) suffices, notice that by (5), the interested quantity can be simplified as

$$\mu^2 \cdot \operatorname{err}(M_{\mu,\varphi}) = \mathbb{E}\|X\|_2^2 \cdot \|\mathcal{I}_\varphi\|_2. \tag{7}$$

Recall that we have $\mathbb{E}X = 0$ by symmetry of $\varphi$ and $\mathbb{E}\nabla\varphi(X) = 0$ by basic Fisher information theory. So $\operatorname{Var}[\nabla\varphi(X)] = \mathbb{E}\|\nabla\varphi(X)\|_2^2 = \operatorname{Tr}\mathbb{E}\nabla\varphi(X)\nabla\varphi(X)^T = \operatorname{Tr}\mathcal{I}_\varphi$. That is, eq. (6) implies

$$\mathbb{E}\|X\|_2^2 \cdot \operatorname{Tr}\mathcal{I}_\varphi \geqslant n^2. \tag{8}$$

Since $\mathcal{I}_\varphi$ is positive semi-definite, by (7) and (8) we have

$$\mu^2 \cdot \operatorname{err}(M_{\mu,\varphi}) = \mathbb{E}\|X\|_2^2 \cdot \|\mathcal{I}_\varphi\|_2 \geqslant \mathbb{E}\|X\|_2^2 \cdot \tfrac{1}{n}\operatorname{Tr}\mathcal{I}_\varphi \geqslant n.$$

Table 1: Explicit expressions of Fisher information and mean squared error.

| Density | $\|\mathcal{I}_\varphi\|_2$ | $\mathbb{E}\|X\|_2^2$ | $\mathbb{E}\|X\|_2^2 \cdot \|\mathcal{I}_\varphi\|_2$ | $\mathbb{E}\|X\|_\infty^2$ | $\mathbb{E}\|X\|_\infty^2 \cdot \|\mathcal{I}_\varphi\|_2$ |
|---|---|---|---|---|---|
| $\propto \mathrm{e}^{-\|x\|_1}$ | $1$ | $2n$ | $2n$ | $\sim (\log n)^2$ | $\sim (\log n)^2$ |
| $\propto \mathrm{e}^{-\|x\|_2}$ | $\frac{1}{n}$ | $n(n+1)$ | $n+1$ | $\sim 2n\log n$ | $\sim 2\log n$ |
| $\propto \mathrm{e}^{-\|x\|_2^2}$ | $2$ | $\frac{1}{2}n$ | $n$ | $\sim \log n$ | $\sim 2\log n$ |
| $\propto \mathrm{e}^{-\|x\|_p^\alpha}$ | Lemma 4.2 | Lemma 4.2 | $\sim C_p \cdot n$ | Appendix | $\leqslant C_p' \cdot (\log n)^{\frac{2}{p}}$ |

Next we focus on the proof of (6). Consider the location family $\{X + \theta : \theta \in \mathbb{R}^n\}$. The Fisher information of this family is $\mathcal{I}_\varphi$ at all $\theta$. The random vector itself is an unbiased estimator of the location. Therefore, by the Cramer–Rao inequality (c.f. [VdV00]), we have that $\mathrm{Cov}(X) - \mathcal{I}_\varphi^{-1}$ is positive semi-definite. As a consequence,

$$\mathrm{Var}[X] = \mathrm{Tr}\,\mathrm{Cov}(X) \geqslant \mathrm{Tr}\,\mathcal{I}_\varphi^{-1} = \lambda_1^{-1} + \cdots + \lambda_n^{-1}.$$

where $\lambda_1 \geqslant \cdots \geqslant \lambda_n > 0$ are the eigenvalues of $\mathcal{I}_\varphi$. We already see that $\mathrm{Var}[\nabla\varphi(X)] = \mathrm{Tr}\,\mathcal{I}_\varphi$, so by Cauchy–Schwarz inequality,

$$\mathrm{Var}[X] \cdot \mathrm{Var}[\nabla\varphi(X)] \geqslant (\lambda_1^{-1} + \cdots + \lambda_n^{-1})(\lambda_1 + \cdots + \lambda_n) \geqslant n^2$$

The proof of the inequality is complete. For standard Gaussian, we have $\mathrm{Cov}(X) = \mathcal{I}_\varphi = I_{n \times n}$, and we have $\mu^2 \cdot \mathrm{err}(M_{\mu,\varphi}) = \mathbb{E}\|X\|_2^2 \cdot \|\mathcal{I}_\varphi\|_2 = \mathrm{Tr}\,I_{n \times n} \cdot 1 = n$. $\qquad\square$

Note that although Theorem 4.1 holds true for very general $\varphi$ (only integrability conditions are imposed in $\mathfrak{F}_n$), the interpretation that $\mu$ is the asymptotic privacy parameter only holds for distributions that satisfy (D1) and (D2). Therefore, let us consider the special case where $\varphi(x) = \|x\|_p^\alpha$. The corresponding $X_\varphi$ will be denoted by $X_{p,\alpha}$ and $\mathcal{I}_\varphi$ by $\mathcal{I}_{p,\alpha}$. In this special case, we can compute the quantities in (8) exactly. In the following lemma, we write $a_n \sim b_n$ for the two sequences $a_n$ and $b_n$ if $\frac{a_n}{b_n} \to 1$ as $n \to \infty$.

**Lemma 4.2.** *For $1 \leqslant p < \infty$ and $1 \leqslant \alpha < \infty$, as $n \to \infty$, we have*

$$\mathbb{E}\|X_{p,\alpha}\|_2^2 \sim n^{\frac{2}{\alpha} - \frac{2}{p} + 1} \cdot \alpha^{-\frac{2}{\alpha}} \cdot p^{\frac{2}{p}} \cdot \Gamma(\tfrac{3}{p}) / \Gamma(\tfrac{1}{p});$$
$$\mathcal{I}_{p,\alpha} \sim n^{\frac{2}{p} - \frac{2}{\alpha}} \cdot \alpha^{\frac{2}{\alpha}} \cdot p^{2 - \frac{2}{p}} \cdot \Gamma(2 - \tfrac{1}{p}) / \Gamma(\tfrac{1}{p}) \cdot I_{n \times n}.$$

This result put Theorem 4.1 into a more concrete context. Some important cases with specific values of $p$ and $\alpha$ are worked out in Table 1. Remarkably, in the last row, the products that characerize privacy-accuracy trade-off are asymptotically independent of $\alpha$. As a by-product of this calculation, we also derive the expression for the isotropic constant of the $n$-dimensional $\ell_p$ ball, which is an important concept in convex geometry (c.f. [BGVV14]). See the appendix for more results and discussion.

Alternatively, we may want to measure the accuracy by the expected squared $\ell_\infty$-norm of the noise. A similar argument suggests to consider the following quantity $\mathbb{E}\|X_\varphi\|_\infty^2 \cdot \|\mathcal{I}_\varphi\|_2$. By Theorem 4.1 and the fact that $\|x\|_\infty \geqslant \frac{1}{\sqrt{n}}\|x\|_2$, we have

$$\mathbb{E}\|X_\varphi\|_\infty^2 \cdot \|\mathcal{I}_\varphi\|_2 \geqslant \frac{1}{n}\mathbb{E}\|X_\varphi\|_2^2 \cdot \|\mathcal{I}_\varphi\|_2 \geqslant 1. \tag{9}$$

We would like to point out a connection to a recently resolved open problem proposed in [SU17], asking if there is a DP algorithm that answers a high-dimensional query with $\ell_2$-sensitivity 1 with $O(1)$ error in $\ell_\infty$ norm. In particular, the recent solution [DK20, GKM20] provides strong evidence that the lower bound in (9) is tight up to a constant factor.

**An Analogy with Uncertainty Principles** There are various mathematical manifestations of the uncertainty principle. The one behind Hesenberg uncertainty principle is that a function and its Fourier transform cannot both be localized simultaneously. Specifically, for a function $f \in L^2(\mathbb{R}^n)$, its Fourier transform is defined as $\hat{f}(\xi) = \int \mathrm{e}^{-2\pi i \langle \xi, x \rangle} f(x)\, \mathrm{d}x$. Fourier transform is unitary, i.e. $\|f\|_{L^2} = \|\hat{f}\|_{L^2}$. In particular, if $|f|^2$ is a probability density, then so is $|\hat{f}|^2$. Our previous

abuse of notation also applies here, for example, $\mathrm{Var}[|f|^2] = \int (x-a)^T (x-a)|f(x)|^2 \, dx$ where $a = \int x|f(x)|^2 \, dx$. For $\|f\|_{L^2} = 1$, we have the following result[5] (c.f. Corollary 2.8 of [FS97])

$$\mathrm{Var}[|f|^2] \cdot \mathrm{Var}[|\hat{f}|^2] \geqslant \tfrac{n^2}{16\pi^2}. \tag{10}$$

The similarity between (6) and (10) suggests that Theorem 4.1 can be considered as yet another manifestation of the uncertainty principle.

## 5 Conclusions and Future Works

In this work, we study constant-sharp optimality of noise addition algorithms for high-dimensional query answering with differential privacy. We demonstrate that the ROC function offers good insight in comparing the "actual spend vs budget" of differential privacy and hence in the design of one-dimensional algorithms. However, when the dimension is high, a CLT shows that $(\varepsilon, \delta)$ privacy budget cannot be fully spent for a large class of noise addition mechanisms as they all behave like a Gaussian mechanism. On the other hand, Fisher information naturally arises in these high-dimensional mechanisms, and the simple and fundamental quantity "privacy parameter × error" automatically manifests itself as the quantity "information × error" in the Cramer–Rao lower bound. Using this, we are able to show an elegant characterization of the precise privacy-accuracy trade-off, and justify the constant-sharp optimality of the Gaussian mechanism. We believe the insights offer a novel perspective to the long-lived privacy-accuracy trade-off question.

Various extensions are possible. An immediate one is to extend the CLT to a broader class of noise distributions, such as log-concave distributions as specified in Conjecture 3.4. Another condition imposed by $\mathfrak{F}_n$ (implicitly) is that the noise must be supported on the whole space. The difficulty in removing the condition lies in the lack of a definition of Fisher information for noise with bounded support. In particular, one may consider extending the theories to cover the noise used in [DK20] and prove a corresponding lower bound like (9). For non-log-concave noise, Lemma 3.5 suggests us to believe that a corresponding log-concave noise with no less privacy and accuracy exists. For algorithms beyond noise addition or problems beyond query answering, we believe that they still exhibit some universal behavior as long as the dimension is high. As a circumstantial evidence, [BDKT12] shows that generic algorithms for query answering can be reduced to a noise addition one with better accuracy and slightly worse privacy.

### Acknowledgements

We thank Jason Hartline, Yin-Tat Lee, Haotian Jiang, Qiyang Han, Sasho Nikolov, Aravindan Vijayaraghavan and Yuansi Chen for helpful comments on earlier versions of the manuscript. J. D. was supported by the NSF HDR TRIPODS award CCF-1934931. W. J. S. was supported in part by NSF through CCF-1763314 and CAREER DMS-1847415, an Alfred Sloan Research Fellowship, and a Facebook Faculty Research Award. L. Z. was supported in part by NSF through DMS-2015378.

### Societal Impact

Private data analysis has positive societal impacts. The major negative concern is that too much utility is sacrificed for privacy. This work is intended to improve our theoretical understanding of such trade-off between privacy and utility.

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
