# Supplemental Materials

In Appendix A we provide the detail of the numerical experiments. The central limit theorem 3.1 (normalized) and 3.2 (general case) are proved in Appendix B. Appendix C proves Lemma 4.2 and provide some additional results that apply beyond norm powers. The proof of Lemma 3.3, which verifies that norm powers satisfy the technical conditions (D1) and (D2), requires results in Appendix C and takes significant effort, so we dedicate the entire Appendix D to it.

## Appendix A   Numerical Verification of the Central Limit Theorem

This section discusses the details of the numerical experiments shown in Figure 1 (repeated below) that verifies our central limit theorem.

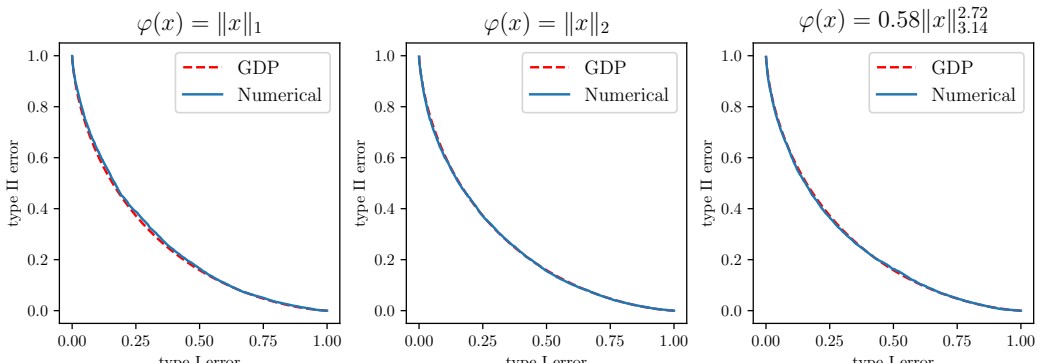

Figure 1: Fast convergence to GDP as claimed in Theorem 1.1. Blue solid curves indicate the true privacy (i.e. ROC functions, see Section 2 for details) of the noise addition mechanism considered in Theorem 1.1. Red dashed curves are GDP limit predicted by our CLT. In all three panels the dimension $n = 30$. In order to show that our theory works for general $\ell_p$-norm to the power $\alpha$, we pick them to be famous mathematical constants, namely $p = \pi, \alpha = \mathrm{e}$ and the coeffcient being the Euler–Mascheroni constant $\gamma$.

The mechanism in consideration is $M(D) = f(D) + tX$ where the $n$-dimensional random vector $X$ has density $\propto \mathrm{e}^{-\|x\|_p^\alpha}$. We want to demonstrate that when $t = \mu^{-1} \cdot \sqrt{\|\mathcal{I}_{p,\alpha}\|_2} = \mu^{-1} \cdot n^{\frac{1}{p} - \frac{1}{\alpha}} \cdot \alpha^{\frac{1}{\alpha}} \cdot p^{1 - \frac{1}{p}} \cdot \sqrt{\frac{\Gamma(2 - \frac{1}{p})}{\Gamma(\frac{1}{p})}} \cdot 1 + (o(1))$ we have

$$\inf_{D, D'} T[M(D), M(D')] \approx G_\mu.$$

The infimum is taken over $D, D'$ such that the direction of $f_n(D) - f_n(D')$ is in a large subset $E_n$ of the unit sphere. It is hard to evaluate the infimum even numerically, but it turns out that the infimum is equal to $\inf_{v \in E_n} T[tX, tX + v]$. This is the first part of the proof of Theorem 1.1.

Therefore, it suffices to evaluate $T(tX, tX + v)$ and compare with the GDP function $G_\mu$, but the high-dimensional nature of $X$ prevents exact evaluation, so we will introduce a Monte Carlo approach.

**Empirical ROC Function**   $X$ has density $\propto \mathrm{e}^{-\varphi(x)}$ and $X + v$ has density $\propto \mathrm{e}^{-\varphi(x-v)}$. The log likelihood ratio is $\varphi(x) - \varphi(x - v)$. Thresholding it at $h$ yields the following type I and type II errors

$$\alpha(h) = \mathbb{P}_{x \sim X}(\varphi(x) - \varphi(x - v) \geqslant h) = \mathbb{P}(\varphi(X) - \varphi(X - v) \geqslant h)$$
$$\beta(h) = \mathbb{P}_{x \sim X+v}(\varphi(x) - \varphi(x - v) < h) = \mathbb{P}(\varphi(X + v) - \varphi(X) < h)$$

Once we have these, $T(X, X + v)$ can be obtained by eliminating $h$ and express $\beta$ as a function of $\alpha$. These two probabilities can be computed by a simple Monte Carlo approach. First We can sample

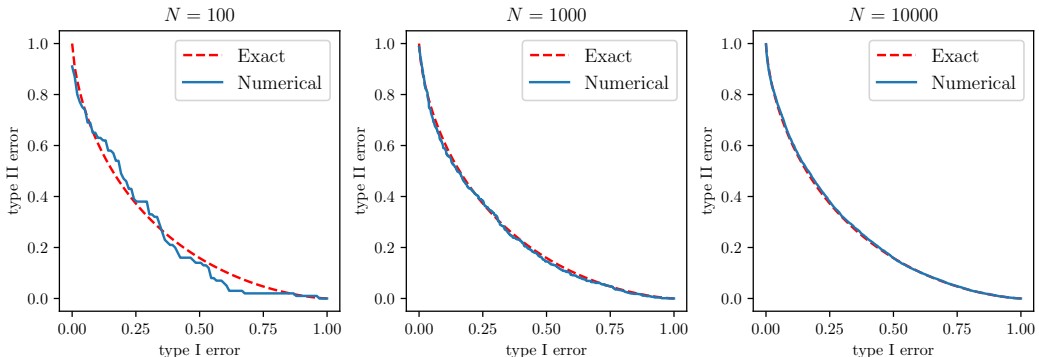

Figure 3: The effect of sample size $N$. We see that $N = 10000$ provides a good estimate. The red dashed curve corresponds to Gaussian noise $\varphi(x) = \frac{1}{2}\|x\|_2^2$, which is why we can perform exact computation. The dimension $n = 30$.

$\{x_1, \ldots, x_N\}$ as i.i.d. copies of $X$. Let

$$a_i = \varphi(x_i - v) - \varphi(x_i)$$
$$b_i = \varphi(x_i + v) - \varphi(x_i)$$
$$\hat{\alpha}(h) = \frac{1}{N} \cdot \#\{a_i \leqslant -h\}$$
$$\hat{\beta}(h) = \frac{1}{N} \cdot \#\{b_i < h\}$$

We only evaluate on a discrete set of $h$ such that the corresponding $\alpha$ forms a uniform grid $\{\frac{1}{N}, \ldots, \frac{N}{N}\}$. Let $h_j = -a_{(j)}$ where $a_{(1)} \leqslant \cdots \leqslant a_{(N)}$ are order statistics of $a_i, i = 1, 2, \ldots, N$. Then for $j = 0, 1, 2, \ldots, N$,

$$\hat{\alpha}_j = \hat{\alpha}(h_j) = \frac{1}{N} \cdot \#\{a_i \leqslant a_{(j)}\} = \frac{j}{N}$$
$$\hat{\beta}_j = \hat{\beta}(h_j) = \frac{1}{N} \cdot \#\{b_i < a_{(j)}\}$$

Let $\hat{T}_N(X, X + v)$ be the function that linearly interpolates the values $\hat{\beta}_0, \ldots, \hat{\beta}_N$ at $\frac{0}{N}, \ldots, \frac{N}{N}$. As a direct consequence of the well-known Glivenko–Cantelli theorem, we have

$$\|\hat{T}_N(X, X + v) - T(X, X + v)\|_\infty \to 0 \text{ almost surely.}$$

We evaluate the effect of the sample size $N$ in Figure 3 and observe that $N = 10000$ works quite well when the $T[X, X + v]$ is close to the 1-GDP function $G_1$ (which is true for all experiments in the paper).

**Evaluating the Fisher Information**   Note that it is numerically infeasible to use the exact expression in Lemma 4.2, since Gamma function grows extremely fast (of course it does, as an interpolation of the factorial). In practice, we find the asymptotic expression in Lemma 4.2 works extremely well.

Next we present the algorithm that samples an $n$-dimensional random vector whose density is $\propto e^{-\|x\|_p^\alpha}$.

It is easy to see its correctness from Lemma C.5 and [CDT98].

## Appendix B   Proof of Theorem 3.1 and 3.2

In this section we first prove the normalized central limit theorem 3.1 and then the general case theorem 3.2. Recall that in normalized CLT, the mechanism in consideration is $M_n(D) = f_n(D) + X_n$, where $X_n$ has density $\propto e^{-\varphi_n}$ with Fisher information $\mathcal{I}_n$ being the identity matrix $I_{n \times n}$.

**Theorem 3.1.** *If the function sequence $\varphi_n$ satisfies conditions (D1) and (D2), then there is a sequence of positive numbers $c_n$ with $c_n \to 0$ as $n \to \infty$ and a subset $E_n \subseteq S^{n-1}$ with $\mathbb{P}_{v \sim S^{n-1}}(v \in E_n) >$*

**Algorithm 1** `Sample` $\propto \mathrm{e}^{-\|x\|_p^\alpha}$

---

1: **Input:** $p, \alpha$ and dimension $n$
  Generate $t \sim \Gamma(\frac{n}{\alpha} + 1, 1)$
  Generate $\xi_i, i = 1, 2, \ldots, n$ i.i.d. from $\Gamma(\frac{1}{p}, 1)$

  Generate random vector $\boldsymbol{x} \in \mathbb{R}^n$ where $x_i = \varepsilon_i \cdot \xi_i^{\frac{1}{p}}$, $\varepsilon_i$ are Rademacher random variables (unbiased coin flips) independent from everything else.
  Generate $r \sim U[0, 1]$
  Let $V = r^{\frac{1}{n}} \cdot \frac{x}{\|x\|_p}$
2: **Output** $t \cdot V$

---

$1 - c_n$ *such that*

$$\| \inf_{D, D'} T[M_n(D), M_n(D')] - G_1 \|_\infty \leqslant c_n$$

*where the infimum is taken over* $D, D'$ *such that* $\frac{f_n(D') - f_n(D)}{\|f_n(D') - f_n(D)\|_2} \in E_n$.

(D1) $\mathrm{KS}(P_v^{\varphi_n}(X_n), v^T \nabla \varphi_n(X_n) = o(1)$ with probability at least $1 - o(1)$ over $v \sim S^{n-1}$
(D2) $\|\nabla \varphi_n(X_n)\|_2 = \sqrt{n} \cdot (1 + o_P(1))$

### B.1 The Main Proof

*Proof of Theorem 3.1.* For clarity, in the proof we drop the subscript $n$ unless the limit $n \to \infty$ is taken. First we show that

$$\inf_{D, D'} T[M(D), M(D')] = \inf_{v \in E_n} T[X, X + v].$$

Notice that

$$T[M(D), M(D')] = T[f(D) + X, f(D') + X] = T[X, X + f(D') - f(D)]$$

Consider the vector $f(D') - f(D)$. Let $v = \frac{f(D') - f(D)}{\|f(D') - f(D)\|_2}$ be its direction and $r$ be its length. We have $f(D') - f(D) = rv$. Since $f$ has $\ell_2$-sensitivity 1, we have $r \in [0, 1]$. The infimum over $D, D'$ can be taken in two steps: first over $v$ and then over $r$. That is,

$$\inf_{D, D'} T[M(D), M(D')] = \inf_{v \in E_n} \inf_{r \in [0, 1]} T[M(D), M(D')] = \inf_{v \in E_n} \inf_{r \in [0, 1]} T[X, X + rv]$$

By Lemma 3.5, $T[X, X + rv]$ is pointwise monotone decreasing in $r$, so for the inner infimum we have $\inf_{r \in [0, 1]} T[X, X + rv] = T[X, X + v]$. Back to the limiting conclusion, it suffices to show that for all $v \in E_n$,

$$\|T[X_n, X_n + v] - G_1\|_\infty \leqslant c_n.$$

To prove this, we use the following lemma

**Lemma B.1.** *Suppose random vector $X$ has density $\propto \mathrm{e}^{-\varphi}$ where $\varphi \in \mathfrak{F}_n$. Let $F_v$ be the CDF of the likelihood projection $P_v^\varphi(X) = \varphi(X + v) - \varphi(X) - \frac{1}{2} v^T \mathcal{I}_\varphi v$, then for any $v \in \mathbb{R}^n$,*

$$T[X, X + v](\alpha) = F_v\big( - F_{-v}^{-1}(\alpha) - v^T \mathcal{I}_\varphi v\big).$$

Let $H_v^n$ and $F_v^n$ be the CDFs of the linear projection $v^T \nabla \varphi_n(X_n)$ and the likelihood projection $P_v^{\varphi_n}(X_n) = \varphi_n(X_n + v) - \varphi_n(X_n) - \frac{1}{2} v^T \mathcal{I}_n v$. When Lemma B.1 is applied to $X_n$ and unit vector $v$, we have

$$T[X_n, X_n + v](\alpha) = F_v^n\big( - (F_{-v}^n)^{-1}(\alpha) - 1\big). \tag{11}$$

Recall that $G_1(\alpha) = \Phi\big( - \Phi^{-1}(\alpha) - 1\big)$ where $\Phi$ is the CDF of standard normal. Comparing with (11), it suffices to show $F_v^n$ is close to $\Phi$. To prove this and to take care of the set $E_n \subseteq S^{n-1}$, we use the two conditions (D1) and (D2), which allow us to apply Sudakov's Theorem (c.f. [Kla10]) stated below as Lemma B.2. In the following, let $\sigma_n$ be the uniform measure (with total measure 1) on the unit sphere $S^{n-1}$.

**Lemma B.2.** *Let $Y_n$ be an isotropic random vector in $\mathbb{R}^n$. Assume that there is $a_n \to 0$ and*

$$\mathbb{P}\left(\left|\frac{\|Y_n\|_2}{\sqrt{n}} - 1\right| \geqslant a_n\right) \leqslant a_n. \tag{12}$$

*Then, there exists $b_n \to 0$ and $\Theta_n \subseteq S^{n-1}$ with $\sigma_n(\Theta_n) \geqslant 1 - b_n$, such that for any $v \in \Theta_n$,*

$$\sup_{t \in \mathbb{R}} |\mathbb{P}(v^T Y_n \leqslant t) - \Phi(t)| \leqslant b_n.$$

Let $Y_n = \varphi_n(X_n)$. $v^T \nabla \varphi_n(X_n) = v^T Y_n$ We know $\mathbb{E} Y_n = \mathbb{E} \nabla \varphi_n(X_n) = 0$, and by the normalization of the Fisher information,

$$\mathbb{E} Y_n Y_n^T = \mathcal{I}_n = I_{n \times n}.$$

Therefore, $Y_n$ is isotropic. Condition (D2) says $\|Y_n\|_2 = \sqrt{n} \cdot (1 + o_P(1))$. That is, $\left|\frac{\|Y_n\|_2}{\sqrt{n}} - 1\right| = o_P(1)$. This implies the existence of $a_n$ in (12). Therefore, we can apply Lemma B.2 to $Y_n = \varphi_n(X_n)$ and conclude that there is $b_n \to 0$ and $\Theta_n \subseteq S^{n-1}$ with $\sigma_n(\Theta_n) \geqslant 1 - b_n$, such that for any $v \in \Theta_n$,

$$\|H_v^n - \Phi\|_\infty \leqslant b_n.$$

Condition (D1) says, there is $b_n' \to 0$ and $\Omega_n \subseteq S^{n-1} \geqslant b_n'$ with $\sigma_n(\Omega_n)$ such that for all $v \in \Omega_n$,

$$\|F_v^n - H_v^n\|_\infty \leqslant b_n'$$

So $v \in \Theta_n \cap \Omega_n$ implies $\|H_v^n - \Phi\|_\infty \leqslant b_n$ and $\|F_v^n - H_v^n\|_\infty \leqslant b_n'$. Therefore, $\|F_v^n - \Phi\|_\infty \leqslant b_n + b_n'$. Set $E_n = \Theta_n \cap (-\Theta_n) \cap \Omega_n \cap (-\Omega_n)$. Then $v \in E_n$ implies

$$\|F_v^n - \Phi\|_\infty \leqslant b_n + b_n' \text{ and } \|F_{-v}^n - \Phi\|_\infty \leqslant b_n + b_n'$$

That is, for any $x \in \mathbb{R}$,

$$\Phi(x) - b_n - b_n' \leqslant F_v^n(x) \leqslant \Phi(x) + b_n + b_n'$$
$$\Phi(x) - b_n - b_n' \leqslant F_{-v}^n(x) \leqslant \Phi(x) + b_n + b_n'$$

As a consequence of the second inequality,

$$\Phi^{-1}(\alpha - b_n - b_n') \leqslant (F_{-v}^n)^{-1}(\alpha) \leqslant \Phi^{-1}(\alpha + b_n + b_n')$$

Therefore, when $v \in \mathbb{E}_n$, by (11) and the inequalities above,

$$
\begin{aligned}
T[X_n, X_n + v](\alpha) &= F_v^n\big(-(F_{-v}^n)^{-1}(\alpha) - 1\big) \\
&\leqslant F_v^n\big(-\Phi^{-1}(\alpha - b_n - b_n') - 1\big) \\
&\leqslant \Phi\big(-\Phi^{-1}(\alpha - b_n - b_n') - 1\big) + b_n + b_n' \\
&= G_1(\alpha - b_n - b_n') + b_n + b_n' \\
&\leqslant G_1(\alpha) + C\sqrt{b_n + b_n'} + b_n + b_n'
\end{aligned}
$$

The final step used the Hölder continuity of $G_\mu$ which we state as Lemma B.3 and prove afterwards.

**Lemma B.3.** *Let $G_\mu = T[N(0,1), N(\mu,1)]$ for $\mu \geqslant 0$. Then $G_\mu$ is $\alpha$-Hölder continuous for any $\alpha < 1$.*

It is worth noting that $G_\mu$ is not 1-Hölder continuous (i.e. Lipschitz continuous) as long as $\mu > 0$.

Without loss of generality assume $C > 1$. Let $c_n = C\sqrt{b_n + b_n'} + b_n + b_n'$. Then $c_n \geqslant 2b_n + 2b_n'$. The above argument shows $T[X_n, X_n + v](\alpha) \leqslant G_1(\alpha) + c_n$. The lower bound can be obtained similarly, so for $v \in E_n$, we have

$$\|T[X_n, X_n + v] - G_1\|_\infty \leqslant c_n.$$

Since all four sets $\Theta_n, -\Theta_n, \Omega_n$ and $-\Omega_n$ are large, we have

$$\sigma_n(E_n) = \sigma_n(\Theta_n \cap (-\Theta_n) \cap \Omega_n \cap (-\Omega_n)) \geqslant 1 - (2b_n + 2b_n') \geqslant 1 - c_n$$

This is the conclusion stated in the theorem. The proof is complete. $\qquad \square$

## B.2 Proof of Lemmas

Next we provide the proofs of the lemmas used, namely Lemmas 3.5, B.1 and B.3.

Let $M^t(D) = f(D) + tX$.

**Lemma 3.5.** *When $X$ has log-concave distribution and $t \geqslant 0$, the ROC function $T[M^t(D), M^t(D')]$ is (pointwise) monotone increasing in $t$ for any $D, D'$.*

*Proof of Lemma 3.5.* Let $0 \leqslant t_1 \leqslant t_2$ and $f_i = T[X, X + t_i v], i = 1, 2$. We need to show $f_1 \geqslant f_2$. Fix $\alpha \in [0, 1]$, let $A_t \subseteq \mathbb{R}^n$ be the optimal rejection region for the testing of $X$ vs $X + tv$. That is,

$$\mathbb{P}[X \in A_t] = \alpha \quad \text{and} \quad \mathbb{P}[X + tv \notin A_t] = T[X, X + tv](\alpha)$$

In order to show $f_1(\alpha) \geqslant f_2(\alpha)$, consider a translated set $A_{t_1} + (t_2 - t_1)v$. This set is at the best suboptimal for the testing of $X$ vs $X + t_2 v$. If we denote $\mathbb{P}[X \in A_{t_1} + (t_2 - t_1)v]$ by $\alpha'$, suboptimality means

$$\mathbb{P}[X + t_2 v \notin A_{t_1} + (t_2 - t_1)v] \geqslant f_2(\alpha').$$

If we can show $\alpha' \leqslant \alpha$, then by the monotonicity of ROC functions, we have

$$\begin{aligned}
f_2(\alpha) &\leqslant f_2(\alpha') \\
&= \mathbb{P}[X + t_2 v \notin A_{t_1} + (t_2 - t_1)v] \\
&= \mathbb{P}[X + t_1 v \notin A_{t_1}] \\
&= f_1(\alpha)
\end{aligned}$$

So the only thing left is to show $\alpha' \leqslant \alpha$, or equivalently,

$$\mathbb{P}[X \in A_{t_1} + (t_2 - t_1)v] \leqslant \mathbb{P}[X \in A_{t_1}].$$

In fact, we will show $A_{t_1} + (t_2 - t_1)v \subseteq A_{t_1}$. This is where log-concavity kicks in. To phrase it more generally, we are going to show that $A_t + sv \subseteq A_t$ for general $t, s \geqslant 0$. Suppose $X$ has density $e^{-\varphi(x)}$ where $\varphi : \mathbb{R}^n \to \mathbb{R} \cup \{+\infty\}$ is a (potentially extended) convex function. By Neyman–Pearson lemma, $A_t = \{x : \varphi(x) - \varphi(x - tv) > h\}$ for some threshold $h$. We would like to show that $x \in A_t$ implies $x + sv \in A_t$. It suffices to show

$$\varphi(x + sv) - \varphi(x + sv - tv) \geqslant \varphi(x) - \varphi(x - tv).$$

In fact, $\varphi(x + sv) - \varphi(x + sv - tv)$ is monotone increasing as a function of $s$. This is a direct consequence of the convexity of $\varphi(x + sv)$ as a function of $s$. More specifically, let $g(s) = \varphi(x + sv)$. Its convexity follows from the convexity of $\varphi$. For $t \geqslant 0$, $\varphi(x + sv) - \varphi(x + sv - tv) = g(s) - g(s - t)$ is easily seen to be monotone by taking a derivative, or from a more rigorous approach following just the definition of convex functions. $\square$

Recall that $\tilde{\varphi}_n(x) = \varphi_n(\frac{x}{t_n})$. The corresponding random vector with density $\propto e^{-\tilde{\varphi}_n}$ is $\tilde{X}_n = X_{\tilde{\varphi}_n}$ and has the same distribution as $t_n X_{\varphi_n}$. The scaling factor normalizes its Fisher information to be an $n \times n$ scalar matrix. In fact,

$$\tilde{\mathcal{I}}_n := \mathcal{I}_{\tilde{\varphi}_n} = t_n^{-2} \mathcal{I}_{\varphi_n} = \mu^2 \cdot I_{n \times n}.$$

*Proof of Lemma B.1.* We are interested in the hypothesis testing $H_0 : X$ vs $H_1 : X + v$. By definition of ROC function in Equation (2), we need to find out the optimal type II error at a given level $\alpha$. By Neymann–Pearson lemma, it suffices to consider likelihood ratio tests. The log density of the null is (up to an additive constant) $-\varphi(x)$, while that of the alternative is $-\varphi(x - v)$. So the log likelihood ratio is $\varphi(x) - \varphi(x - v)$. Under null it is distributed as $\varphi(X) - \varphi(X - v)$ and thresholding at $h$ yields type I error

$$\begin{aligned}
\alpha &= \mathbb{P}(\varphi(X) - \varphi(X - v) > h) \\
&= \mathbb{P}(\varphi(X - v) - \varphi(X) < -h) \\
&= \mathbb{P}(\varphi(X - v) - \varphi(X) - \tfrac{1}{2}v^T \mathcal{I}_\varphi v < -h - \tfrac{1}{2}v^T \mathcal{I}_\varphi v) \\
&= F_{-v}(-h - \tfrac{1}{2}v^T \mathcal{I}_\varphi v)
\end{aligned}$$

Under the alternative, the log likelihood ratio is distributed as $\varphi(X+v)-\varphi(X)$, so the corresponding type II error is

$$\begin{aligned}
\beta &= \mathbb{P}(\varphi(X+v)-\varphi(X) < h) \\
&= \mathbb{P}(\varphi(X+v)-\varphi(X) - \tfrac{1}{2}v^T\mathcal{I}_\varphi v < hh - \tfrac{1}{2}v^T\mathcal{I}_\varphi v) \\
&= F_v(h - \tfrac{1}{2}v^T\mathcal{I}_\varphi v)
\end{aligned}$$

From the expression of $\alpha$ we can solve for $h$:

$$h = -F_{-v}^{-1}(\alpha) - \tfrac{1}{2}v^T\mathcal{I}v$$

Plugging this into the expression of $\beta$ yields

$$\begin{aligned}
\beta &= F_v(h - \tfrac{1}{2}v^T\mathcal{I}_\varphi v) \\
&= F_v\big(-F_{-v}^{-1}(\alpha) - v^T\mathcal{I}v\big)
\end{aligned}$$

The ROC function maps $\alpha$ to the minimal $\beta$, so this is exactly the expression of $T[X, X+v]$. $\qquad\square$

*Proof of Lemma B.3.* We know that $G_\mu(x) = \Phi\big(-\Phi^{-1}(x)-\mu\big)$. It suffices to show that $1-G_\mu(x) = \Phi\big(\Phi^{-1}(x)+\mu\big)$ is $\alpha$-Hölder continuous for any $\alpha < 1$.

**Lemma B.4.** *Consider $f, g : [0,1] \to \mathbb{R}$. Suppose $f, g$ and $g - f$ is monotone increasing and $g$ is $\alpha$-Hölder continuous, then $f$ is also $\alpha$-Hölder continuous.*

To see this, notice that by monotonicity of $g - f$, we have $g(x) - f(x) \leqslant g(y) - f(y)$ for $x < y$. Hence

$$|f(y) - f(x)| = f(y) - f(x) \leqslant g(y) - g(x) \leqslant Cx^\alpha.$$

**Lemma B.5.** *For each $\alpha < 1$, there is an $\varepsilon = \varepsilon(\alpha) > 0$ such that $x^\alpha - (1 - G_\mu(x))$ is monotone increasing in $[0, \varepsilon]$.*

Let $\alpha = 1 - \delta$. $h(x) = x^\alpha - (1 - G_\mu(x)) = x^\alpha - \Phi\big(\Phi^{-1}(x) + \mu\big)$. Let $y = \Phi^{-1}(x)$. Then $x = \Phi(y)$

$$\begin{aligned}
h'(x) &= \alpha x^{\alpha-1} - \frac{\mathrm{d}}{\mathrm{d}x}\Phi(y + \mu) \\
&= \alpha x^{-\delta} - \phi(y + \mu) \cdot \frac{\mathrm{d}y}{\mathrm{d}x} \\
&= \alpha x^{-\delta} - \phi(y + \mu)\Big/\frac{\mathrm{d}x}{\mathrm{d}y} \\
&= \alpha x^{-\delta} - \frac{\phi(y + \mu)}{\phi(y)} \\
&= \alpha x^{-\delta} - \mathrm{e}^{-\mu y - \frac{1}{2}\mu^2} \\
&= \mathrm{e}^{\delta \log x^{-1} + \log \alpha} - \mathrm{e}^{-\mu y - \frac{1}{2}\mu^2}
\end{aligned}$$

It is known that $|\Phi^{-1}(x)| = -\Phi^{-1}(x) \leqslant \sqrt{2\log x^{-1}}$. So for fixed $\alpha, \delta$ and $\mu$, there is an $\varepsilon$ such that when $x \in [0, \varepsilon]$, we have

$$\delta \log x^{-1} + \log \alpha \geqslant \mu\Phi^{-1}(x) - \frac{1}{2}\mu^2 = -\mu y - \frac{1}{2}\mu^2$$

Hence,

$$h'(x) = \mathrm{e}^{\delta \log x^{-1} + \log \alpha} - \mathrm{e}^{-\mu y - \frac{1}{2}\mu^2} \geqslant 0$$

$\qquad\square$

Interestingly, this implies the following result:

**Proposition B.6.** *For each $\alpha \in [0, 1)$, there is a $C > 0$ such that*

$$\int_{a+1}^{b+1} e^{-x^2} \, dx \leqslant C \left( \int_a^b e^{-x^2} \, dx \right)^\alpha.$$

For a convex $\varphi$ such that $e^{-\varphi}$ is integrable, let $F_v^\varphi$ be the cdf of $P_v^\varphi(X_\varphi)$. Dropping the unnecessary subscripts and superscripts of $\varphi$, we have

## Appendix C  Proof of Lemma 4.2

The major goal of this section is the following extended version of Lemma 4.2. We proceed by first presenting some general results in Appendix C.1, followed by calculation for norm powers in Appendix C.2.

**Lemma C.1.** *For $1 \leqslant p < \infty$ and $1 \leqslant \alpha < \infty$, as $n \to \infty$, we have*

$$\mathbb{E}\|X_{p,\alpha}\|_2^2 = \frac{\Gamma(\frac{n}{\alpha} + 1 + \frac{2}{\alpha})}{\Gamma(\frac{n}{p} + 1 + \frac{2}{p})} \cdot \frac{\Gamma(\frac{n}{p} + 1)}{\Gamma(\frac{n}{\alpha} + 1)} \cdot n \cdot \frac{\Gamma(\frac{3}{p})}{\Gamma(\frac{1}{p})}$$

$$\sim n^{\frac{2}{\alpha} - \frac{2}{p} + 1} \cdot \alpha^{-\frac{2}{\alpha}} \cdot p^{\frac{2}{p}} \cdot \frac{\Gamma(\frac{3}{p})}{\Gamma(\frac{1}{p})}$$

$$\mathcal{I}_{p,\alpha} = \alpha^2 \cdot \frac{\Gamma(\frac{n+2\alpha-2}{\alpha})}{\Gamma(\frac{n+2p-2}{p})} \cdot \frac{\Gamma(\frac{n}{p})}{\Gamma(\frac{n}{\alpha})} \cdot \frac{\Gamma(2 - \frac{1}{p})}{\Gamma(\frac{1}{p})} \cdot I_{n \times n}$$

$$\sim n^{\frac{2}{p} - \frac{2}{\alpha}} \cdot \alpha^{\frac{2}{\alpha}} \cdot p^{2 - \frac{2}{p}} \cdot \frac{\Gamma(2 - \frac{1}{p})}{\Gamma(\frac{1}{p})} \cdot I_{n \times n}.$$

### C.1  Regarding Homogeneous $\varphi$

In this section, in addition to that $\varphi \in \mathfrak{F}_n$, we further assume that $\varphi : \mathbb{R}^n \to \mathbb{R}$ is positively homogeneous. Recall that $\varphi$ is (positively) homogeneous of degree $\alpha > 0$ if $\varphi(tx) = |t|^\alpha \varphi(x)$ for $t \in \mathbb{R}, x \in \mathbb{R}^n$. This implies $\varphi(0) = 0$.

The first result takes care of the normalizer $Z_\varphi$ defined as $\int e^{-\varphi(x)} \, dx$.

**Lemma C.2.** $Z_\varphi = \Gamma(\frac{n}{\alpha} + 1) \cdot \text{vol}(K_\varphi)$ where $K_\varphi = \{x : \varphi(x) \leqslant 1\}$.

*Proof of Lemma C.2.* We use polar coordinate. For any function $f : \mathbb{R}^n \to \mathbb{R}$,

$$\int_{\mathbb{R}^n} f(x) \, dx = \int_0^\infty \int_{S^{n-1}} f(r\theta) r^{n-1} \, d\theta \, dr$$

So

$$Z = \int_{\mathbb{R}^n} e^{-\varphi(x)} \, dx = \int_{S^{n-1}} \int_0^\infty e^{-\varphi(r\theta)} r^{n-1} \, dr \, d\theta$$

$$= \int_{S^{n-1}} \int_0^\infty e^{-r^\alpha \varphi(\theta)} r^{n-1} \, dr \, d\theta$$

Let $t = r^\alpha, r = t^{\frac{1}{\alpha}}, dr = \frac{1}{\alpha} t^{\frac{1}{\alpha} - 1} \, dt$.

$$\int_0^\infty e^{-r^\alpha \varphi(\theta)} r^{n-1} \, dr = \int_0^\infty e^{-t\varphi(\theta)} \cdot t^{\frac{n-1}{\alpha}} \cdot \frac{1}{\alpha} t^{\frac{1}{\alpha} - 1} \, dt$$

$$= \frac{1}{\alpha} \int_0^\infty e^{-t\varphi(\theta)} \cdot t^{\frac{n}{\alpha} - 1} \, dt$$

$$= \frac{1}{\alpha} \cdot \frac{\Gamma(\frac{n}{\alpha})}{\varphi(\theta)^{\frac{n}{\alpha}}}$$

So
$$Z = \int_{S^{n-1}} \frac{1}{\alpha} \cdot \frac{\Gamma(\frac{n}{\alpha})}{\varphi(\theta)^{\frac{n}{\alpha}}} \, d\theta$$
On the other hand, consider a set defined with polar coordinate:
$$K := \{(r, \theta) : r \leqslant \rho(\theta)\}.$$

Its volume is
$$\text{vol}(K) = \int_{\mathbb{R}^n} 1_K(x) \, dx$$
$$= \int_{S^{n-1}} \int_0^{\rho(\theta)} r^{n-1} \, dr \, d\theta$$
$$= \frac{1}{n} \int_{S^{n-1}} \rho(\theta)^n \, d\theta$$

We see that
$$Z = \int_{S^{n-1}} \frac{1}{\alpha} \cdot \frac{\Gamma(\frac{n}{\alpha})}{\varphi(\theta)^{\frac{n}{\alpha}}} \, d\theta$$
$$= \frac{\Gamma(\frac{n}{\alpha})}{\alpha} \cdot n \cdot \frac{1}{n} \int_{S^{n-1}} \left[ \varphi(\theta)^{-\frac{1}{\alpha}} \right]^n \, d\theta$$
$$= \tfrac{n}{\alpha} \cdot \Gamma(\tfrac{n}{\alpha}) \cdot \text{vol}(K_\varphi)$$

where
$$K_\varphi = \{(r, \theta) : r \leqslant \varphi(\theta)^{-\frac{1}{\alpha}}\} = \{(r, \theta) : r^\alpha \varphi(\theta) \leqslant 1\} = \{(r, \theta) : \varphi(r\theta) \leqslant 1\} = \{x : \varphi(x) \leqslant 1\}.$$
Noticing $\Gamma(z+1) = z\Gamma(z)$, we have
$$Z = \Gamma(\tfrac{n}{\alpha} + 1) \cdot \text{vol}(K_\varphi)$$

$\square$

**Lemma C.3.** *The $m$-th moment of a $\Gamma(k, 1)$ distribution is $\frac{\Gamma(m+k)}{\Gamma(k)}$.*

*Proof of Lemma C.3.* The $m$-th moment of a $\Gamma(k, 1)$ distribution is
$$\frac{1}{\Gamma(k)} \int_0^\infty x^m \cdot x^{k-1} e^{-x} \, dx = \frac{\Gamma(m+k)}{\Gamma(k)}.$$

$\square$

The following result also appears in [Wan05].
**Lemma C.4.**
$$\text{vol}(K_p) = 2^n \cdot \frac{\Gamma(\frac{1}{p} + 1)^n}{\Gamma(\frac{n}{p} + 1)}.$$

*Proof of Lemma C.4.* By Lemma C.2, we have
$$\text{vol}(K_p) = \frac{1}{\Gamma(\frac{n}{p} + 1)} \cdot \int e^{-\sum |x_i|^p} \, dx$$
$$= \frac{1}{\Gamma(\frac{n}{p} + 1)} \cdot \left( \int_{-\infty}^{+\infty} e^{-|x|^p} \, dx \right)^n$$
$$\int_{-\infty}^{+\infty} e^{-|x|^p} \, dx = 2 \int_0^{+\infty} e^{-x^p} \, dx$$
$$= 2 \int_0^{+\infty} e^{-y} \, dy^{\frac{1}{p}}$$
$$= \frac{2}{p} \int_0^{+\infty} e^{-y} y^{\frac{1}{p} - 1} \, dy$$
$$= \tfrac{2}{p} \Gamma(\tfrac{1}{p}) = 2\Gamma(\tfrac{1}{p} + 1)$$

$\square$

**Lemma C.5.** *Let* $t \sim \Gamma(\frac{n}{\alpha} + 1, 1)$. *Let* $V_\varphi$ *has uniform distribution over* $K_\varphi$ *where* $K_\varphi = \{x : \varphi(x) \leqslant 1\}$ *independently from* $t$. *Then* $t^{\frac{1}{\alpha}} \cdot V$ *has density* $\frac{1}{Z} e^{-\varphi}$.

*Proof of Lemma C.5.* We use a more principled way: assume $r$ has density $p(r)$ over $(0, +\infty)$ and $rV$ has density $\frac{1}{Z} e^{-\varphi}$, find $p(r)$. Let $B$ be a small ball.

$$\mathbb{P}(rV \in x + B) = \int_0^\infty \mathbb{P}(V \in \frac{x + B}{r}) \cdot p(r) \, \mathrm{d}r$$

$$\mathbb{P}(V \in \frac{x + B}{r}) = \begin{cases} 0, & \text{if } \frac{x}{r} \notin K_\varphi \\ \dfrac{\mathrm{vol}(\frac{B}{r})}{\mathrm{vol}(K_\varphi)}, & \text{if } \frac{x}{r} \in K_\varphi. \end{cases}$$

$\frac{x}{r} \in K_\varphi \Leftrightarrow \varphi(\frac{x}{r}) \leqslant 1 \Leftrightarrow \varphi(x) \leqslant r^\alpha \Leftrightarrow r \geqslant \varphi(x)^{1/\alpha}$. So

$$\mathbb{P}(rV \in x + B) = \int_0^\infty \mathbb{P}(V \in \frac{x + B}{r}) \cdot p(r) \, \mathrm{d}r$$

$$= \int_{\varphi(x)^{1/\alpha}}^\infty \frac{\mathrm{vol}(\frac{B}{r})}{\mathrm{vol}(K_\varphi)} \cdot p(r) \, \mathrm{d}r$$

$$= \mathrm{vol}(B) \cdot \int_{\varphi(x)^{1/\alpha}}^\infty \frac{p(r) r^{-n}}{\mathrm{vol}(K_\varphi)} \, \mathrm{d}r.$$

So the density of $rV$ at $x$ is $\int_{\varphi(x)^{1/\alpha}}^\infty \frac{p(r) r^{-n}}{\mathrm{vol}(K_\varphi)} \, \mathrm{d}r$. In order to match it with $\frac{1}{Z} e^{-\varphi}$, we have

$$\int_{\varphi(x)^{1/\alpha}}^\infty \frac{p(r) r^{-n}}{\mathrm{vol}(K_\varphi)} \, \mathrm{d}r = \frac{e^{-\varphi(x)}}{\Gamma(\frac{n}{\alpha} + 1) \cdot \mathrm{vol}(K_\varphi)}$$

$$\int_{\varphi(x)^{1/\alpha}}^\infty p(r) r^{-n} \, \mathrm{d}r = \frac{e^{-\varphi(x)}}{\Gamma(\frac{n}{\alpha} + 1)}$$

Let $\varphi(x)^{1/\alpha} = u$, we have

$$\int_u^\infty p(r) r^{-n} \, \mathrm{d}r = \frac{1}{\Gamma(\frac{n}{\alpha} + 1)} \cdot e^{-u^\alpha}.$$

Taking derivative with respect to $u$, we have

$$p(u) u^{-n} = \frac{1}{\Gamma(\frac{n}{\alpha} + 1)} \cdot e^{-u^\alpha} \cdot \alpha u^{\alpha - 1}.$$

It's straightforward to show that if $t \sim \Gamma(\frac{n}{\alpha} + 1, 1)$, then $t^{\frac{1}{\alpha}}$ has the above density $p(u)$. $\square$

A simple but useful corollary of Lemma C.5 is

**Corollary C.6.** *Let* $t \sim \Gamma(\frac{n}{\alpha} + 1, 1)$. *Let* $U$ *has uniform distribution over* $\partial K_\varphi$, *independently from* $t$ *and* $r$ *with density* $nx^{n-1}$ *over* $[0, 1]$, *independent from* $t$ *and* $U$. *Then* $t^{\frac{1}{\alpha}} \cdot r \cdot U$ *has density* $\frac{1}{Z} e^{-\varphi}$.

We commented that we can compute the isotropic constants for $\ell_p$ balls. The rest of the section is dedicated to this kind of results.

Let $\mu$ be a log-concave probability measure on $\mathbb{R}^n$ with density $f_\mu : \mathbb{R}^n \to \mathbb{R}_{\geqslant 0}$. The isotropic constant of $\mu$ is defined by (see e.g. [Gia])

$$L_\mu = \left( \sup_{x \in \mathbb{R}^n} f_\mu(x) \right)^{\frac{1}{n}} \cdot \left( \det \mathrm{Cov}(\mu) \right)^{\frac{1}{2n}}.$$

As a special case, when $\mu$ is the uniform distribution over the convex body $K$, the corresponding isotropic constant is denoted by $L_K$ and has expression

$$L_K = \mathrm{vol}(K)^{-\frac{1}{n}} \cdot \left( \det \mathrm{Cov}(\mu) \right)^{\frac{1}{2n}}.$$

For homogeneous and convex $\varphi : \mathbb{R}^n \to \mathbb{R}$, we use $L_\varphi$ to denote the isotropic constant of its associated probability distribution, i.e. the one with density $\frac{1}{Z_\varphi} e^{-\varphi(x)}$. With the help of Lemma C.5, we can relate $L_\varphi$ to the isotropic constant of its unit ball $L_{K_\varphi}$.

**Lemma C.7.**
$$L_\varphi = \frac{[\Gamma(\frac{n}{\alpha} + 1 + \frac{2}{\alpha})]^{\frac{1}{2}}}{[\Gamma(\frac{n}{\alpha} + 1)]^{\frac{1}{2} + \frac{1}{n}}} \cdot L_{K_\varphi}$$

*Proof.*
$$\text{Cov}(X_\varphi) = \mathbb{E}[X_\varphi X_\varphi^T] = \mathbb{E}[t^{\frac{2}{\alpha}} \cdot V_\varphi V_\varphi^T] = \mathbb{E}t^{\frac{2}{\alpha}} \cdot \mathbb{E}[V_\varphi V_\varphi^T] = \mathbb{E}t^{\frac{2}{\alpha}} \cdot \text{Cov}(V_\varphi)$$

$$\det \text{Cov}(X_\varphi) = (\mathbb{E}t^{\frac{2}{\alpha}})^n \cdot \det \text{Cov}(V_\varphi)$$

$$\begin{aligned}
L_\varphi &= Z_\varphi^{-\frac{1}{n}} \cdot (\det \text{Cov}(X_\varphi))^{\frac{1}{2n}} \\
&= Z_\varphi^{-\frac{1}{n}} \cdot (\mathbb{E}t^{\frac{2}{\alpha}})^{\frac{1}{2}} \cdot (\det \text{Cov}(V_\varphi))^{\frac{1}{2n}} \\
&= \Gamma(\tfrac{n}{\alpha} + 1)^{-\frac{1}{n}} \cdot \text{vol}(K_\varphi)^{-\frac{1}{n}} \cdot (\mathbb{E}t^{\frac{2}{\alpha}})^{\frac{1}{2}} \cdot (\det \text{Cov}(V_\varphi))^{\frac{1}{2n}} \qquad \text{(Lemma C.2)} \\
&= (\mathbb{E}t^{\frac{2}{\alpha}})^{\frac{1}{2}} \cdot \Gamma(\tfrac{n}{\alpha} + 1)^{-\frac{1}{n}} \cdot L_{K_\varphi}
\end{aligned}$$

By Lemma C.3, $\mathbb{E}t^{\frac{2}{\alpha}} = \frac{\Gamma(\frac{n}{\alpha} + 1 + \frac{2}{\alpha})}{\Gamma(\frac{n}{\alpha} + 1)}$. So

$$\begin{aligned}
L_\varphi &= \left(\frac{\Gamma(\frac{n}{\alpha} + 1 + \frac{2}{\alpha})}{\Gamma(\frac{n}{\alpha} + 1)}\right)^{\frac{1}{2}} \cdot \Gamma(\tfrac{n}{\alpha} + 1)^{-\frac{1}{n}} \cdot L_{K_\varphi} \\
&= \frac{[\Gamma(\frac{n}{\alpha} + 1 + \frac{2}{\alpha})]^{\frac{1}{2}}}{[\Gamma(\frac{n}{\alpha} + 1)]^{\frac{1}{2} + \frac{1}{n}}} \cdot L_{K_\varphi}
\end{aligned}$$

$\square$

The last result is a sufficient condition of the Fisher information being a scalar matrix.

**Lemma C.8.** *If $\varphi$ is invariant under the action of $\{\pm 1\}^n$ and cyclic group of size $n$, i.e.*

1. $\varphi(\pm x_1, \pm x_2, \ldots, \pm x_n) = \varphi(x_1, x_2, \ldots, x_n)$

2. $\varphi(x_1, x_2, \ldots, x_{n-1}, x_n) = \varphi(x_2, x_3, \ldots, x_n, x_1)$

*then $\mathcal{I}_\varphi = \frac{1}{n}\mathbb{E}\|\nabla\varphi\|_2^2 \cdot I$.*

*Proof of Lemma C.8.* First we use the symmetry to show $\mathcal{I}_\varphi = cI$ for some $c$.
$$\begin{aligned}
\varphi(x_1, x_2, \ldots, x_n) &= \varphi(-x_1, x_2, \ldots, x_n) \\
\partial_1\varphi(x_1, x_2, \ldots, x_n) &= -\partial_1\varphi(-x_1, x_2, \ldots, x_n) \\
\partial_2\varphi(x_1, x_2, \ldots, x_n) &= \partial_2\varphi(-x_1, x_2, \ldots, x_n)
\end{aligned}$$

This shows $\partial_1\varphi \cdot \partial_2\varphi$ is an odd function of $x_1$. On the other hand, we know the density $\mathrm{e}^{-\varphi}$ is an even function of $x_1$. So we conclude that $\mathbb{E}[\partial_1\varphi \cdot \partial_2\varphi] = 0$. Similarly, we can show that for any $i \neq j$, $\mathbb{E}[\partial_i\varphi \cdot \partial_j\varphi] = 0$. This shows that $\mathcal{I}_\varphi$ is a diagonal matrix.

By cyclic symmetry, we have

$$\begin{aligned}
\varphi(x_1, x_2, \ldots, x_{n-1}, x_n) &= \varphi(x_2, x_3, \ldots, x_n, x_1) \\
\partial_1\varphi(x_1, x_2, \ldots, x_{n-1}, x_n) &= \partial_n\varphi(x_2, x_3, \ldots, x_n, x_1) \\
\partial_1\varphi(x_1, x_2, \ldots, x_{n-1}, x_n)^2 \mathrm{e}^{-\varphi(x_1,x_2,\ldots,x_{n-1},x_n)} &= \partial_n\varphi(x_2, x_3, \ldots, x_n, x_1)^2 \mathrm{e}^{-\varphi(x_1,x_2,\ldots,x_{n-1},x_n)} \\
&= \partial_n\varphi(y_1, y_2, \ldots, y_{n-1}, y_n)^2 \mathrm{e}^{-\varphi(y_n,y_1,\ldots,y_{n-1})} \\
&= \partial_n\varphi(y_1, y_2, \ldots, y_{n-1}, y_n)^2 \mathrm{e}^{-\varphi(y_1,y_2,\ldots,y_n)}
\end{aligned}$$

This shows $\mathbb{E}\partial_1\varphi^2 = \cdots = \mathbb{E}\partial_n\varphi^2$. Hence $\mathcal{I}_\varphi = cI$ for some $c$.

$$\text{Tr}\,\mathcal{I}_\varphi = \text{Tr}\,\mathbb{E}[\nabla\varphi\nabla\varphi^T] = \mathbb{E}[\text{Tr}\,\nabla\varphi\nabla\varphi^T] = \mathbb{E}[\text{Tr}\,\nabla\varphi^T\nabla\varphi] = \mathbb{E}\|\nabla\varphi\|_2^2.$$

On the other hand, $\text{Tr}\,\mathcal{I}_\varphi = \text{Tr}\,cI = cn$, so $c = \frac{1}{n}\mathbb{E}\|\nabla\varphi\|_2^2$.

$\square$

## C.2 Calculation for Norm Powers

**Lemma 4.2.** *For $1 \leqslant p < \infty$ and $1 \leqslant \alpha < \infty$, as $n \to \infty$, we have*

$$\mathbb{E}\|X_{p,\alpha}\|_2^2 \sim n^{\frac{2}{\alpha} - \frac{2}{p} + 1} \cdot \alpha^{-\frac{2}{\alpha}} \cdot p^{\frac{2}{p}} \cdot \Gamma(\tfrac{3}{p})/\Gamma(\tfrac{1}{p});$$

$$\mathcal{I}_{p,\alpha} \sim n^{\frac{2}{p} - \frac{2}{\alpha}} \cdot \alpha^{\frac{2}{\alpha}} \cdot p^{2 - \frac{2}{p}} \cdot \Gamma(2 - \tfrac{1}{p})/\Gamma(\tfrac{1}{p}) \cdot I_{n \times n}.$$

We divide the proof into two parts, one for each of the equations.

*Proof of Lemma 4.2 (variance part).* By Lemma C.5, let $t \sim \Gamma(\frac{n}{\alpha} + 1, 1)$ random variable and $V_p$ has uniform distribution over the $\ell_p$ unit ball $K_p$.

$$\mathbb{E}\|X_{p,\alpha}\|_2^2 = \mathbb{E}t^{\frac{2}{\alpha}} \cdot \mathbb{E}\|V_p\|_2^2 = \frac{\Gamma(\frac{n}{\alpha} + 1 + \frac{2}{\alpha})}{\Gamma(\frac{n}{\alpha} + 1)} \cdot \mathbb{E}\|V_p\|_2^2 \qquad (13)$$

Setting $\alpha = p$ yields

$$\mathbb{E}\|X_{p,p}\|_2^2 = \frac{\Gamma(\frac{n}{p} + 1 + \frac{2}{p})}{\Gamma(\frac{n}{p} + 1)} \cdot \mathbb{E}\|V_p\|_2^2 \qquad (14)$$

The reason we do this is that $\mathbb{E}\|X_{p,p}\|_2^2$ can be computed explicitly. In fact,

$$\mathbb{E}\|X_{p,p}\|_2^2 = \frac{1}{Z_n} \int \sum x_i^2 \cdot e^{-\sum |x_i|^p} \, dx$$

$$= \frac{n}{Z_n} \int x_1^2 \cdot e^{-\sum |x_i|^p} \, dx$$

$$= \frac{n}{Z_n} \int x_1^2 \cdot e^{-|x_1|^p} \, dx_1 \cdot Z_{n-1}$$

where $Z_n = \int e^{-\sum |x_i|^p} \, dx$. We know by Lemma C.4 that

$$Z_n = \Gamma(\tfrac{n}{p} + 1) \cdot \text{vol}(K_p) = 2^n \cdot \Gamma(\tfrac{1}{p} + 1)^n$$

and

$$\int_{-\infty}^{+\infty} x^2 \cdot e^{-|x|^p} \, dx = 2 \int_0^\infty x^2 \cdot e^{-x^p} \, dx = 2 \int_0^\infty y^{\frac{2}{p}} \cdot e^{-y} \cdot \frac{1}{p} y^{\frac{1}{p} - 1} \, dy = \frac{2}{p} \int_0^\infty y^{\frac{3}{p} - 1} \cdot e^{-y} \, dy = \frac{2}{p} \cdot \Gamma(\tfrac{3}{p})$$

So

$$\mathbb{E}\|X_{p,p}\|_2^2 = \frac{n}{Z_n} \int x_1^2 \cdot e^{-|x_1|^p} \, dx_1 \cdot Z_{n-1}$$

$$= n \cdot \frac{2}{p} \cdot \Gamma(\tfrac{3}{p}) \cdot \frac{Z_{n-1}}{Z_n}$$

$$= \frac{n \cdot \frac{2}{p} \cdot \Gamma(\tfrac{3}{p})}{2\Gamma(\tfrac{1}{p} + 1)}$$

$$= \frac{n \cdot \frac{2}{p} \cdot \Gamma(\tfrac{3}{p})}{\frac{2}{p}\Gamma(\tfrac{1}{p})} = n \cdot \frac{\Gamma(\tfrac{3}{p})}{\Gamma(\tfrac{1}{p})}$$

Plugging this into (14), we have

$$\mathbb{E}\|V_p\|_2^2 = \frac{\Gamma(\frac{n}{p} + 1)}{\Gamma(\frac{n}{p} + 1 + \frac{2}{p})} \cdot \mathbb{E}\|X_{p,p}\|_2^2$$

$$= \frac{\Gamma(\frac{n}{p} + 1)}{\Gamma(\frac{n}{p} + 1 + \frac{2}{p})} \cdot n \cdot \frac{\Gamma(\tfrac{3}{p})}{\Gamma(\tfrac{1}{p})}.$$

Using this in (13),

$$\mathbb{E}\|X_{p,\alpha}\|_2^2 = \frac{\Gamma(\frac{n}{\alpha}+1+\frac{2}{\alpha})}{\Gamma(\frac{n}{\alpha}+1)} \cdot \mathbb{E}\|V_p\|_2^2$$

$$= \frac{\Gamma(\frac{n}{\alpha}+1+\frac{2}{\alpha})}{\Gamma(\frac{n}{\alpha}+1)} \cdot \frac{\Gamma(\frac{n}{p}+1)}{\Gamma(\frac{n}{p}+1+\frac{2}{p})} \cdot n \cdot \frac{\Gamma(\frac{3}{p})}{\Gamma(\frac{1}{p})}$$

In order to study the asymptotics of $\mathbb{E}\|X_{p,\alpha}\|_2^2$ as $n \to \infty$, recall Stirling's formula

$$\Gamma(z+1) \sim \sqrt{2\pi z}\left(\frac{z}{e}\right)^z.$$

So we have

$$\frac{\Gamma(\frac{n}{\alpha}+1+\frac{2}{\alpha})}{\Gamma(\frac{n}{\alpha}+1)} \sim \frac{\sqrt{\frac{n+2}{\alpha}} \cdot \left(\frac{n+2}{\alpha e}\right)^{\frac{n+2}{\alpha}}}{\sqrt{\frac{n}{\alpha}} \cdot \left(\frac{n}{\alpha e}\right)^{\frac{n}{\alpha}}}$$

$$\sim \left(\frac{n+2}{\alpha e} \cdot \frac{\alpha e}{n}\right)^{\frac{n}{\alpha}} \cdot \left(\frac{n+2}{\alpha e}\right)^{\frac{2}{\alpha}}$$

$$= \left(1+\frac{2}{n}\right)^{\frac{n}{2} \cdot \frac{2}{\alpha}} \cdot \left(\frac{n+2}{\alpha e}\right)^{\frac{2}{\alpha}}$$

$$\sim \left(\frac{n+2}{\alpha}\right)^{\frac{2}{\alpha}}$$

Hence

$$\mathbb{E}\|X_{p,\alpha}\|_2^2 \sim \left(\frac{n+2}{\alpha}\right)^{\frac{2}{\alpha}} \Big/ \left(\frac{n+2}{p}\right)^{\frac{2}{p}} \cdot n \cdot \frac{\Gamma(\frac{3}{p})}{\Gamma(\frac{1}{p})} \sim n^{\frac{2}{\alpha}-\frac{2}{p}+1} \cdot \alpha^{-\frac{2}{\alpha}} \cdot p^{\frac{2}{p}} \cdot \frac{\Gamma(\frac{3}{p})}{\Gamma(\frac{1}{p})}$$

$$\square$$

Before we proceed to the proof of the Fisher information part of Lemma 4.2, we derive the isotropic constants results as promised, using Lemma C.7 and the variance part of Lemma 4.2.

**Corollary C.9.** *The isotropic constant of $n$-dimensional $\ell_p$ ball is*

$$L_{K_p}^2 = \frac{p^2}{4} \cdot \frac{\Gamma(\frac{3}{p})}{[\Gamma(\frac{1}{p})]^3} \cdot \frac{[\Gamma(\frac{n}{p}+1)]^{1+\frac{2}{n}}}{\Gamma(\frac{n}{p}+1+\frac{2}{p})}$$

*Proof of Corollary C.9.* For a general convex body $K$, let $V_K$ be a random vector with the uniform distribution over $K$. Recall that the isotropic constant of $K$ is

$$L_K = \mathrm{vol}(K)^{-\frac{1}{n}} \cdot \left(\det \mathrm{Cov}(V_K)\right)^{\frac{1}{2n}}.$$

Now we focus on the unit ball of $\ell_p$ norm $K_p$. The corresponding random vector is denoted by $V_p$ By a symmetry argument similar to Lemma C.8, we have that

$$\mathrm{Cov}(V_p) = \frac{1}{n} \cdot \mathbb{E}\|V_p\|_2^2 \cdot I_{n \times n}.$$

Combining this and Lemma C.4, we have

$$L_{K_p}^2 = \text{vol}(K_p)^{-\frac{2}{n}} \cdot \frac{1}{n}\mathbb{E}\|V_p\|_2^2$$

$$= \left(2^n \cdot \frac{\Gamma(\frac{1}{p}+1)^n}{\Gamma(\frac{n}{p}+1)}\right)^{-\frac{2}{n}} \cdot \frac{\Gamma(\frac{n}{p}+1)}{\Gamma(\frac{n}{p}+1+\frac{2}{p})} \cdot \frac{\Gamma(\frac{3}{p})}{\Gamma(\frac{1}{p})}$$

$$= \frac{1}{4} \cdot \frac{\Gamma(\frac{n}{p}+1)^{\frac{2}{n}}}{\Gamma(\frac{1}{p}+1)^2} \cdot \frac{\Gamma(\frac{n}{p}+1)}{\Gamma(\frac{n}{p}+1+\frac{2}{p})} \cdot \frac{\Gamma(\frac{3}{p})}{\Gamma(\frac{1}{p})}$$

$$= \frac{1}{4} \cdot \frac{\Gamma(\frac{n}{p}+1)^{\frac{2}{n}}}{\frac{1}{p^2}\Gamma(\frac{1}{p})^2} \cdot \frac{\Gamma(\frac{n}{p}+1)}{\Gamma(\frac{n}{p}+1+\frac{2}{p})} \cdot \frac{\Gamma(\frac{3}{p})}{\Gamma(\frac{1}{p})}$$

$$= \frac{p^2}{4} \cdot \frac{\Gamma(\frac{3}{p})}{[\Gamma(\frac{1}{p})]^3} \cdot \frac{[\Gamma(\frac{n}{p}+1)]^{1+\frac{2}{n}}}{\Gamma(\frac{n}{p}+1+\frac{2}{p})}$$

$\square$

**Corollary C.10.** *When* $\varphi(x) = \|x\|_p^\alpha$,

$$L_{p,\alpha}^2 = \frac{p^2}{4} \cdot \frac{\Gamma(\frac{3}{p})}{[\Gamma(\frac{1}{p})]^3} \cdot \frac{\Gamma(\frac{n}{\alpha}+1+\frac{2}{\alpha})}{\Gamma(\frac{n}{p}+1+\frac{2}{p})} \cdot \left(\frac{\Gamma(\frac{n}{p}+1)}{\Gamma(\frac{n}{\alpha}+1)}\right)^{1+\frac{2}{n}}$$

*Proof of Corollary C.10.* Directly follows from the above result and Lemma C.7. $\square$

Now we turn our attention back to the proof of Lemma 4.2.

*Proof of Lemma 4.2 (variance part).* By Lemma C.8, $\mathcal{I}_{p,\alpha} = \|\mathcal{I}_{p,\alpha}\|_2 \cdot I_{n \times n}$ and $\|\mathcal{I}_{p,\alpha}\|_2 = \frac{1}{n}\mathbb{E}\|\nabla\varphi\|_2^2$ where $\varphi(x) = \|x\|_p^\alpha$. In this case the gradient has an explicit expression:

$$[\nabla\varphi(x)]_i = \alpha \cdot \left(\sum |x_i|^p\right)^{\frac{\alpha}{p}-1} \cdot |x_i|^{p-1} \cdot \text{sgn}(x_i) = \alpha\|x\|_p^{\alpha-p} \cdot |x_i|^{p-1} \cdot \text{sgn}(x_i)$$

$$\|\nabla\varphi(x)\|_2^2 = \alpha^2 \sum |x_i|^{2p-2} \cdot \left(\sum |x_i|^p\right)^{\frac{2\alpha}{p}-2} = \alpha^2\|x\|_{2p-2}^{2p-2} \cdot \|x\|_p^{2\alpha-2p}$$

By Corollary C.6, $X_{p,\alpha} \stackrel{d}{=} t^{\frac{1}{\alpha}} \cdot r \cdot U_p$, where $t \sim \Gamma(\frac{n}{\alpha}+1, 1)$, $U$ has uniform distribution over $\partial K_p$ and $r$ has density $nx^{n-1}$ over $[0, 1]$.

$$\|\nabla\varphi\|_2^2 = \alpha^2\|t^{\frac{1}{\alpha}}rU\|_{2p-2}^{2p-2} \cdot \|t^{\frac{1}{\alpha}}rU\|_p^{2\alpha-2p}$$

$$= \alpha^2 t^{\frac{2\alpha-2}{\alpha}} r^{2\alpha-2} \cdot \|U\|_{2p-2}^{2p-2} \cdot \|U\|_p^{2\alpha-2p}$$

Since $t, r, U_p$ are independent and $\|U_p\|_p = 1$, we have

$$\|\mathcal{I}_{p,\alpha}\|_2 = \frac{1}{n}\mathbb{E}\|\nabla\varphi\|_2^2 = \frac{\alpha^2}{n} \cdot \mathbb{E}t^{\frac{2\alpha-2}{\alpha}} \cdot \mathbb{E}r^{2\alpha-2} \cdot \mathbb{E}\|U\|_{2p-2}^{2p-2} \tag{15}$$

By Lemma C.3,

$$\mathbb{E}t^{\frac{2\alpha-2}{\alpha}} = \frac{\Gamma(\frac{n}{\alpha}+1+\frac{2\alpha-2}{\alpha})}{\Gamma(\frac{n}{\alpha}+1)} = \frac{\Gamma(\frac{n+2\alpha-2}{\alpha}+1)}{\Gamma(\frac{n}{\alpha}+1)}.$$

The moment of $r$ can be computed directly

$$\mathbb{E}r^{2\alpha-2} = \int_0^1 r^{2\alpha-2} \cdot nr^{n-1}\,\mathrm{d}r = \frac{n}{n+2\alpha-2}.$$

Plugging into (15), we have

$$\|\mathcal{I}_{p,\alpha}\|_2 = \frac{\alpha^2}{n} \cdot \mathbb{E}t^{\frac{2\alpha-2}{\alpha}} \cdot \mathbb{E}r^{2\alpha-2} \cdot \mathbb{E}\|U_p\|_{2p-2}^{2p-2}$$

$$= \frac{\alpha^2}{n} \cdot \frac{\Gamma(\frac{n+2\alpha-2}{\alpha}+1)}{\Gamma(\frac{n}{\alpha}+1)} \cdot \frac{n}{n+2\alpha-2} \cdot \mathbb{E}\|U_p\|_{2p-2}^{2p-2}$$

$$= \frac{\alpha^2}{n} \cdot \frac{\frac{n+2\alpha-2}{\alpha} \cdot \Gamma(\frac{n+2\alpha-2}{\alpha})}{\frac{n}{\alpha} \cdot \Gamma(\frac{n}{\alpha})} \cdot \frac{n}{n+2\alpha-2} \cdot \mathbb{E}\|U_p\|_{2p-2}^{2p-2}$$

$$= \frac{\alpha^2}{n} \cdot \frac{\Gamma(\frac{n+2\alpha-2}{\alpha})}{\Gamma(\frac{n}{\alpha})} \cdot \mathbb{E}\|U_p\|_{2p-2}^{2p-2} \qquad (*)$$

On the other hand, when $p = \alpha$, we have

$$\|\nabla\varphi(x)\|_2^2 = p^2 \sum |x_i|^{2p-2}.$$

In this case, $x$ has joint density $\propto e^{-\|x\|_p^p} = e^{-\sum |x_i|^p}$. Let $Y_1, Y_2, \ldots, Y_n$ be i.i.d. random variables with density $\propto e^{-|y|^p}$. Then we have

$$\|\mathcal{I}_{p,p}\|_2 = \frac{1}{n}\mathbb{E}\|\nabla\varphi\|_2^2 = \frac{1}{n} \cdot p^2 \cdot \mathbb{E}\sum |Y_i|^{2p-2} = p^2\mathbb{E}\big[|Y_i|^{2p-2}\big]$$

Let $z = \int_{-\infty}^{+\infty} e^{-|y|^p}\,\mathrm{d}y$.

$$\mathbb{E}\big[|Y_i|^{2p-2}\big] = \frac{1}{z}\int_{-\infty}^{+\infty} |y|^{2p-2} \cdot e^{-|y|^p}\,\mathrm{d}y$$

$$= \frac{\int_0^{+\infty} y^{2p-2} \cdot e^{-y^p}\,\mathrm{d}y}{\int_0^{+\infty} e^{-y^p}\,\mathrm{d}y}$$

$$= \frac{\int_0^{+\infty} x^{\frac{2p-2}{p}} \cdot x^{\frac{1}{p}-1} \cdot e^{-x}\,\mathrm{d}x}{\int_0^{+\infty} x^{\frac{1}{p}-1} \cdot e^{-x}\,\mathrm{d}x}$$

$$= \frac{\Gamma(2-\frac{1}{p})}{\Gamma(\frac{1}{p})}$$

Relating to $(*)$ in the special case of $\alpha = p$,

$$\frac{p^2}{n} \cdot \frac{\Gamma(\frac{n+2p-2}{p})}{\Gamma(\frac{n}{p})} \cdot \mathbb{E}\|U_p\|_{2p-2}^{2p-2} = \|\mathcal{I}_{p,p}\|_2 = p^2 \cdot \frac{\Gamma(2-\frac{1}{p})}{\Gamma(\frac{1}{p})}$$

Hence

$$\mathbb{E}\|U_p\|_{2p-2}^{2p-2} = \frac{n}{p^2} \cdot \frac{\Gamma(\frac{n}{p})}{\Gamma(\frac{n+2p-2}{p})} \cdot \|\mathcal{I}_{p,p}\|_2 = n \cdot \frac{\Gamma(\frac{n}{p})}{\Gamma(\frac{n+2p-2}{p})} \cdot \frac{\Gamma(2-\frac{1}{p})}{\Gamma(\frac{1}{p})}$$

Using $(*)$ again,

$$\|\mathcal{I}_{p,\alpha}\|_2 = \frac{\alpha^2}{n} \cdot \frac{\Gamma(\frac{n+2\alpha-2}{\alpha})}{\Gamma(\frac{n}{\alpha})} \cdot \mathbb{E}\|U_p\|_{2p-2}^{2p-2}$$

$$= \frac{\alpha^2}{n} \cdot \frac{\Gamma(\frac{n+2\alpha-2}{\alpha})}{\Gamma(\frac{n}{\alpha})} \cdot n \cdot \frac{\Gamma(\frac{n}{p})}{\Gamma(\frac{n+2p-2}{p})} \cdot \frac{\Gamma(2-\frac{1}{p})}{\Gamma(\frac{1}{p})}$$

$$= \alpha^2 \cdot \frac{\Gamma(\frac{n+2\alpha-2}{\alpha})}{\Gamma(\frac{n}{\alpha})} \cdot \frac{\Gamma(\frac{n}{p})}{\Gamma(\frac{n+2p-2}{p})} \cdot \frac{\Gamma(2-\frac{1}{p})}{\Gamma(\frac{1}{p})}$$

In order to study the asymptotics, using Stirling's formula again,

$$
\frac{\Gamma\left(\frac{2\alpha-2+n}{\alpha}\right)}{\Gamma(\frac{n}{\alpha}+1)} \sim \frac{\sqrt{\frac{n+\alpha-2}{\alpha}} \cdot \left(\frac{n+\alpha-2}{\alpha e}\right)^{\frac{n+\alpha-2}{\alpha}}}{\sqrt{\frac{n}{\alpha}} \cdot \left(\frac{n}{\alpha e}\right)^{\frac{n}{\alpha}}}
$$

$$
\sim \left(\frac{n+\alpha-2}{\alpha e} \cdot \frac{\alpha e}{n}\right)^{\frac{n}{\alpha}} \cdot \left(\frac{n+\alpha-2}{\alpha e}\right)^{\frac{\alpha-2}{\alpha}}
$$

$$
= \left(1+\frac{\alpha-2}{n}\right)^{\frac{n}{\alpha}} \cdot \left(\frac{n+\alpha-2}{\alpha e}\right)^{\frac{\alpha-2}{\alpha}}
$$

$$
\sim e^{\frac{\alpha-2}{\alpha}} \cdot \left(\frac{n+\alpha-2}{\alpha e}\right)^{\frac{\alpha-2}{\alpha}}
$$

$$
\sim \left(\frac{n+\alpha-2}{\alpha}\right)^{\frac{\alpha-2}{\alpha}}
$$

$$
\sim n^{1-\frac{2}{\alpha}} \cdot \alpha^{-1+\frac{2}{\alpha}}
$$

$$
\|\mathcal{I}_{p,\alpha}\|_2 = \alpha^2 \cdot \frac{\Gamma(\frac{n+2\alpha-2}{\alpha})}{\Gamma(\frac{n}{\alpha})} \cdot \frac{\Gamma(\frac{n}{p})}{\Gamma(\frac{n+2p-2}{p})} \cdot \frac{\Gamma(2-\frac{1}{p})}{\Gamma(\frac{1}{p})}
$$

$$
= \alpha^2 \cdot \frac{\Gamma(\frac{n+2\alpha-2}{\alpha})}{\Gamma(\frac{n}{\alpha}+1)} \cdot \frac{n}{\alpha} \cdot \frac{\Gamma(\frac{n}{p})}{\Gamma(\frac{n+2p-2}{p})} \cdot \frac{p}{n} \cdot \frac{\Gamma(2-\frac{1}{p})}{\Gamma(\frac{1}{p})}
$$

$$
\sim \alpha p \cdot n^{1-\frac{2}{\alpha}} \cdot \alpha^{-1+\frac{2}{\alpha}} \cdot n^{-1+\frac{2}{p}} \cdot p^{1-\frac{2}{p}} \cdot \frac{\Gamma(2-\frac{1}{p})}{\Gamma(\frac{1}{p})}
$$

$$
\sim \alpha^{\frac{2}{\alpha}} \cdot p^{2-\frac{2}{p}} \cdot \frac{\Gamma(2-\frac{1}{p})}{\Gamma(\frac{1}{p})} \cdot n^{\frac{2}{p}-\frac{2}{\alpha}}
$$

This finishes the entire proof of Lemma 4.2. $\qquad\qquad\square$

## Appendix D  Proof of Lemma 3.3

Without the loss of generality, we assume $\mu = 1$ in the proof.

By Lemma 4.2, we then have $t_n = \frac{1}{\mu} \cdot \sqrt{\|\mathcal{I}_{\varphi_n}\|_2} \asymp n^{\frac{1}{\alpha}-\frac{1}{p}}$, so the rescaled $\tilde{\varphi}_n : \mathbb{R}^n \to \mathbb{R}$ has the form

$$
\tilde{\varphi}_n(x) = c_{p,\alpha} n^{1-\frac{\alpha}{p}} \|x\|_p^\alpha,
$$

where $c_{p,\alpha} = \alpha^{-1} \cdot p^{-\alpha+\frac{\alpha}{p}} \cdot \left(\frac{\Gamma(2-\frac{1}{p})}{\Gamma(\frac{1}{p})}\right)^{-\frac{\alpha}{2}}$.

### D.1  Lemmas

We first state a few auxiliary lemmas.

**Lemma D.1.**

$$
\|X\|_p = \left((\frac{1}{\alpha})^{1/\alpha} + o_P(1)\right) \cdot n^{\frac{1}{p}}.
$$

**Lemma D.2.** *If* $p > 1, \alpha > 0$*, then*

$$
\sum_{i:|X_i|\leqslant 2|v_i|} (|X_i + v_i|^p - |X_i|^p) = o_P(1).
$$

$$
\sum_{i:|X_i|\leqslant 2|v_i|} p\,\mathrm{sgn}(X_i)|X_i|^{p-1}v_i = o_P(1).
$$

$$
\sum_{i:|X_i|<2|v_i|} \frac{p(p-1)}{2}|X_i|^{p-2}v_i^2 = o_P(1)
$$

### D.2 Main proof

We will first prove the case where $p > 1$. The proof of the corner case where $p = 1$ will be given in Section D.5

**Verification of Condition D1.**

Recall that

$$\tilde{\varphi}_n(x) = c_{p,\alpha} n^{1-\frac{\alpha}{p}} \|x\|_p^\alpha,$$

where $c_{p,\alpha} = \alpha^{-1} \cdot p^{-\alpha+\frac{\alpha}{p}} \cdot \left(\frac{\Gamma(2-\frac{1}{p})}{\Gamma(\frac{1}{p})}\right)^{-\frac{\alpha}{2}}$.

We then have

$$P_v^\varphi(X) = \varphi(X+v) - \varphi(X) - \frac{1}{2} v^T \mathcal{I}_\varphi v$$

$$= c_{p,\alpha} n^{1-\frac{\alpha}{p}} (\|X+v\|_p^\alpha - \|X\|_p^\alpha - \frac{1}{2})$$

$$\nabla\phi(X) = c_{p,\alpha} n^{1-\frac{\alpha}{p}} \cdot \alpha \|X\|_p^{\alpha-p} \cdot \mathrm{sgn}(X) \odot X^{\odot(p-1)}$$

Now let us consider

$$P_v^\varphi(X) - v^T \nabla\varphi(X) = c_{p,\alpha} n^{1-\frac{\alpha}{p}} (\|X+v\|_p^\alpha - \|X\|_p^\alpha - \frac{1}{2} - \langle \alpha \|X\|_p^{\alpha-p} \cdot \mathrm{sgn}(X) \odot X^{\odot(p-1)}, v\rangle).$$

Then

$$\|X+v\|_p^\alpha - \|X\|_p^\alpha - \langle \alpha \|X\|_p^{\alpha-p} \cdot \mathrm{sgn}(X) \odot X^{\odot(p-1)}, v\rangle$$

$$\asymp (1 + o_P(1)) \frac{\alpha \|X\|_p^{\alpha-p}}{p} \left(\|X+v\|_p^p - \|X\|_p^p - p\,\mathrm{sgn}(X) \odot X^{\odot(p-1)} \odot v\right)$$

$$\asymp \frac{1}{p} \cdot \alpha^{\frac{p}{\alpha}} \cdot n^{\frac{\alpha-p}{p}} \left(\|X+v\|_p^p - \|X\|_p^p - p\,\mathrm{sgn}(X) \odot \cdot X^{\odot(p-1)} \odot v\right)$$

$$\asymp \frac{1}{p} \cdot \alpha^{\frac{p}{\alpha}} \cdot n^{\frac{\alpha-p}{p}} \left(\sum_{i=1}^n (|X_i+v_i|^p - |X_i|^p - p\,\mathrm{sgn}(X_i) \cdot X_i^{p-1})v_i\right)$$

To prove (1), it suffices to show $\left(\sum_{i=1}^n (|X_i+v_i|^p - |X_i|^p) - p\,\mathrm{sgn}(X_i) \cdot X_i^{p-1} v_i - \frac{1}{2}\right) = o_P(1)$.
We expand this expression as

$$\sum_{i=1}^n (|X_i+v_i|^p - |X_i|^p - p\,\mathrm{sgn}(X_i) \cdot X_i^{p-1} v_i) = \sum_{i:|X_i|>2|v_i|} (|X_i+v_i|^p - |X_i|^p - p\,\mathrm{sgn}(X_i) \cdot X_i^{p-1} v_i)$$

$$+ \sum_{i:|X_i|\leqslant 2|v_i|} (|X_i+v_i|^p - |X_i|^p)$$

$$- \sum_{i:|X_i|\leqslant 2|v_i|} p\,\mathrm{sgn}(X_i)|X_i|^{p-1} v_i$$

When $|X_i| > 2|v_i|$, note that

$$|X_i+v_i|^p = |X_i|^p(1 + v_i/X_i)^p = |X_i|^p \left(1 + p\frac{v_i}{X_i} + \frac{p(p-1)}{2}\frac{v_i^2}{X_i^2} + O\left(\frac{v_i^3}{X_i^3}\right)\right).$$

Combing with Lemma D.2, we then have

$$\sum_{i=1}^n (|X_i+v_i|^p - |X_i|^p - p\,\mathrm{sgn}(X_i) \cdot X_i^{p-1} v_i)$$

$$= \sum_{i:|X_i|>2|v_i|} |X_i|^p \left(\frac{p(p-1)}{2}\frac{v_i^2}{X_i^2} + O\left(\frac{v_i^3}{X_i^3}\right)\right) + o_P(1)$$

In order to verify Condition D1, we need to show: 1). $\sum_{i:|X_i|>2|v_i|} \frac{p(p-1)}{2}|X_i|^{p-2}v_i^2$ converges to a constant; 2). Show the third order term is vanishing, that is, $\sum_i |X_i|^{p-3}|v_i|^3 \to 0$.

We will prove the proposition in the following two steps:

**Step 1.**

We need to prove that

$$\sum_{i:|X_i|>2|v_i|} \frac{p(p-1)}{2}|X_i|^{p-2}v_i^2$$

converges to a constant.

This can be computed similarly as before. By Lemma D.2

$$\sum_{i:|X_i|<2|v_i|} \frac{p(p-1)}{2}|X_i|^{p-2}v_i^2 = o_P(1)$$

Therefore, we have

$$\sum_{i:|X_i|>2|v_i|} \frac{p(p-1)}{2}|X_i|^{p-2}v_i^2 = \sum_i \frac{p(p-1)}{2}|X_i|^{p-2}v_i^2 + o_P(1).$$

We further obtain

$$\sum_i \frac{p(p-1)}{2}|X_i|^{p-2}v_i^2 = \frac{p(p-1)}{2} \sum_i v_i^2 \cdot \left(\frac{n}{\alpha}\right)^{\frac{p-2}{\alpha}} \cdot |U_i|^{p-2}.$$

Again, we also have

$$\sum_{i=1}^n v_i^2 |\tilde{X}_i|^{p-2} \sim \left(\frac{n}{p}\right)^{\frac{p-2}{p}} \cdot \sum_{i=1}^n v_i^2 |\tilde{U}_i|^{p-2}$$

$$= \left(\frac{n}{p}\right)^{\frac{p-2}{p}} \cdot \sum_{i=1}^n v_i^2 |U_i|^{p-2} \cdot n^{\frac{p-2}{\alpha} - \frac{p-2}{p}}$$

$$= n^{\frac{p-2}{\alpha}} \cdot p^{\frac{2}{p}-1} \cdot \sum_{i=1}^n v_i^2 |U_i|^{p-2}.$$

This implies

$$\sum_{i=1}^n v_i^2 |U_i|^{p-2} = n^{-\frac{p-2}{\alpha}} \cdot p^{1-\frac{2}{p}} \sum_i v_i^2 |\tilde{X}_i|^{p-2} \xrightarrow{p} n^{-\frac{p-2}{\alpha}} \cdot p^{1-\frac{2}{p}} \cdot \frac{\Gamma(1-\frac{1}{p})}{\Gamma(\frac{1}{p})}.$$

Then we have

$$\sum_i \frac{p(p-1)}{2}|X_i|^{p-2}v_i^2 = \frac{p(p-1)}{2} \cdot \left(\frac{n}{\alpha}\right)^{\frac{p-2}{\alpha}} \sum_i v_i^2 \cdot |U_i|^{p-2}$$

$$\xrightarrow{p} \frac{1}{2} \cdot \alpha^{-\frac{p-2}{\alpha}} \cdot p^{3-\frac{2}{p}} \cdot \frac{\Gamma(2-\frac{1}{p})}{\Gamma(\frac{1}{p})}.$$

**Step 2.** Show the third order term is vanishing

Prove that

$$\sum_i |X_i|^{p-3}|v_i|^3 \to 0.$$

$$\sum_i |X_i|^{p-3}|v_i|^3 \leqslant n \cdot \left(\sqrt{\frac{\log n}{n}}\right)^3 \cdot \max_i |X_i|^{p-3}$$

According to Theorem C.5, we have

$$X \stackrel{d}{=} t^{1/\alpha} \cdot r \cdot U,$$

where $t \sim \Gamma(\frac{n}{\alpha} + 1, 1)$, $U$ has uniform distribution over $\partial K_\varphi = \{x : \|x\|_p = n^{1/p-1/\alpha}\}$, $r$ has density $nx^{n-1}$ over $[0, 1]$, and $U, r, t$ are independent.

As a result, by Corollary C.6

$$\max_i |X_i| \sim (\frac{n}{\alpha})^{\frac{1}{\alpha}} \cdot \max_i |U_i|.$$

Consider $\tilde{X} \sim e^{-\|x\|_p^p}$, we then have

$$\tilde{X} \stackrel{d}{=} \tilde{t} \cdot r \cdot \tilde{U},$$

where $\tilde{t} \sim \Gamma(\frac{n}{p} + 1, 1)$ and $\tilde{U} = n^{1/\alpha - 1/p} U$.

Therefore,

$$\begin{aligned}
\max_i |\tilde{X}_i| &\sim (\frac{n}{p})^{\frac{1}{p}} \cdot \max_i |\tilde{U}_i| \\
&= (\frac{n}{p})^{\frac{1}{p}} \cdot \max_i |U_i| \cdot n^{\frac{1}{\alpha} - \frac{1}{p}} \\
&= n^{\frac{1}{\alpha}} \cdot p^{-\frac{1}{p}} \cdot \max_i |U_i|.
\end{aligned}$$

Therefore, we have

$$\max_i |U_i| \sim n^{-\frac{1}{\alpha}} \cdot p^{\frac{1}{p}} \max_i |\tilde{X}_i| \lesssim n^{-\frac{1}{\alpha}} \cdot p^{\frac{1}{p}} (\log n)^{1/p},$$

and we have

$$\max_i |X_i| \sim (\frac{n}{\alpha})^{\frac{1}{\alpha}} \cdot \max_i |U_i| \lesssim (\log n)^{1/p}$$

As a result,

$$\sum_i |X_i|^{p-3} |v_i|^3 \leqslant n \cdot (\sqrt{\frac{\log n}{n}})^3 \cdot \max_i |X_i|^{p-3} = (\log n)^{2.5 - 3/p} \cdot n^{-1/2} \to 0.$$

**Verification of Condition (D2).** Use Sudakov's theorem to prove asymptotic normality.

Note that by Lemma D.2

$$\sum_{i : |X_i| > 2|v_i|} p \operatorname{sgn}(X_i) |X_i|^{p-1} v_i = o_P(1) + \sum_{i=1}^n p \operatorname{sgn}(X_i) |X_i|^{p-1} v_i.$$

It suffices to show $\sum_{i=1}^n p \operatorname{sgn}(X_i) |X_i|^{p-1} v_i$ is asymptotically a normal random variable.

Firstly, we have

$$\sum_{i=1}^n (p \operatorname{sgn}(X_i) |X_i|^{p-1})^2 = p^2 \sum_{i=1}^n |X_i|^{2(p-1)}$$

According to Theorem C.5, we have

$$X \stackrel{d}{=} t^{1/\alpha} \cdot r \cdot U,$$

where $t \sim \Gamma(\frac{n}{\alpha} + 1, 1)$, $U$ has uniform distribution over $\partial K_\varphi = \{x : \|x\|_p = n^{1/p-1/\alpha}\}$, $r$ has density $nx^{n-1}$ over $[0, 1]$, and $U, r, t$ are independent.

As a result, by Corollary C.6

$$\sum_{i=1}^n |X_i|^{2(p-1)} \sim (\frac{n}{\alpha})^{\frac{2p-2}{\alpha}} \cdot \sum_{i=1}^n |U_i|^{2p-2}.$$

Consider $\tilde{X} \sim e^{-\|x\|_p^p}$, we then have

$$\tilde{X} \overset{d}{=} \tilde{t} \cdot r \cdot \tilde{U},$$

where $\tilde{t} \sim \Gamma(\frac{n}{p} + 1, 1)$ and $\tilde{U} = n^{1/\alpha - 1/p} U$.

Therefore,

$$\sum_{i=1}^{n} |\tilde{X}_i|^{2(p-1)} \sim (\frac{n}{p})^{\frac{2p-2}{p}} \cdot \sum_{i=1}^{n} |\tilde{U}_i|^{2p-2}$$

$$= (\frac{n}{p})^{\frac{2p-2}{p}} \cdot \sum_{i=1}^{n} |U_i|^{2p-2} \cdot n^{\frac{2p-2}{\alpha} - \frac{2p-2}{p}}$$

$$= n^{\frac{2p-2}{\alpha}} \cdot p^{\frac{2}{p}-2} \cdot \sum_{i=1}^{n} |U_i|^{2p-2}.$$

This implies

$$\sum_{i=1}^{n} |U_i|^{2p-2} = n^{-\frac{2p-2}{\alpha}} \cdot p^{2-2/p} \sum_{i=1}^{n} |\tilde{X}_i|^{2(p-1)} \overset{p}{\to} p^{2-2/p} n^{1-\frac{2p-2}{\alpha}} \cdot \mathbb{E}[|\tilde{X}_i|^{2p-2}],$$

where

$$\mathbb{E}[|\tilde{X}_i|^{2p-2}] = \frac{1}{\frac{2}{p}\Gamma(\frac{1}{p})} \int_{-\infty}^{\infty} |x|^{2p-2} e^{-|x|^p} \, dx = 2 \int_{0}^{\infty} x^{2p-2} e^{-x^p} \, dx = \frac{\Gamma(2 - \frac{1}{p})}{\Gamma(\frac{1}{p})}$$

is a constant when $p$ is of constant order.

As a result, let $C = p^{4-2/p} \cdot \alpha^{-\frac{2p-2}{\alpha}} \cdot \frac{\Gamma(2-\frac{1}{p})}{\Gamma(\frac{1}{p})}$, we have

$$\sum_{i=1}^{n} (p \operatorname{sgn}(X_i)|X_i|^{p-1})^2 = p^2 \sum_{i=1}^{n} |X_i|^{2(p-1)} \sim p^2 (\frac{n}{\alpha})^{\frac{2p-2}{\alpha}} \cdot \sum_{i=1}^{n} |U_i|^{2p-2} \overset{p}{\to} Cn,$$

satisfying the thin-shell condition of Sudakov's theorem and therefore $\sum_{i=1}^{n} p \operatorname{sgn}(X_i)|X_i|^{p-1} v_i$ is asymptotically normal with variance $Cn$.

### D.3   Proof of Lemma 10.1

According to Theorem C.5, we have

$$X \overset{d}{=} t^{1/\alpha} \cdot r \cdot U,$$

where $t \sim \Gamma(\frac{n}{\alpha} + 1, 1)$, $U$ has uniform distribution over $\partial K_\varphi = \{x : \|x\|_p = n^{1/p-1/\alpha}\}$, $r$ has density $nx^{n-1}$ over $[0, 1]$, and $U, r, t$ are independent.

As a result,

$$\|X\|_p = |t^{1/\alpha}| \cdot |r| \cdot n^{1/p-1/\alpha} \sim (\frac{1}{\alpha})^{1/\alpha} \cdot n^{1/p}$$

### D.4   Proof of Lemma 10.2

Denote $z_i = I(|X_i| \leqslant 2|v_i|) = I(|t^{1/\alpha} \cdot r \cdot U_i| \leqslant 2|v_i|) \leqslant I(|t^{1/\alpha} \cdot r \cdot U_i| \leqslant 2\sqrt{\frac{2\log n}{n}})$.

Consider $\tilde{X} \sim e^{-\|x\|_p^p}$, we then have

$$\tilde{X} \overset{d}{=} \tilde{t}^{1/p} \cdot r \cdot \tilde{U},$$

where $\tilde{t} \sim \Gamma(\frac{n}{p} + 1, 1)$ and $\tilde{U} = n^{1/\alpha - 1/p} U$.

Since $p/n = o(1)$, we then have $\tilde{t} \sim \frac{\alpha}{p} \cdot t$ and $\tilde{t}^{1/p} \sim \frac{n^{1/p-1/\alpha} \cdot \alpha^{1/\alpha}}{p^{1/p}} \cdot t^{1/\alpha}$, then

$$\frac{1}{n}\sum_{i=1}^{n} I(|t^{1/\alpha} \cdot r \cdot U_i| \leqslant 2\sqrt{\frac{2\log n}{n}}) = \frac{1}{n}\sum_{i=1}^{n} I(|\tilde{t}^{1/p} \cdot r \cdot \tilde{U}_i| \leqslant 2\frac{\alpha^{1/\alpha}}{p^{1/p}}\sqrt{\frac{2\log n}{n}})$$

$$= \frac{1}{n}\sum_{i=1}^{n} I(|\tilde{X}_i| \leqslant 2\frac{n^{1/p-1/\alpha} \cdot \alpha^{1/\alpha}}{p^{1/p}}\sqrt{\frac{2\log n}{n}})$$

$$\sim \mathbb{P}(|\tilde{X}_i| \leqslant 2\frac{\alpha^{1/\alpha}}{p^{1/p}}\sqrt{\frac{2\log n}{n}}),$$

where $\tilde{X}_i$ are *i.i.d.* drawn from the population with density $\propto e^{-|x|^p}$.

When $|x| \leqslant 2\frac{\alpha^{1/\alpha}}{p^{1/p}}\sqrt{\frac{2\log n}{n}}$, and $2\frac{\alpha^{1/\alpha}}{p^{1/p}}\sqrt{\frac{2\log n}{n}} = o(1)$, we then have

$$\mathbb{P}(|\tilde{X}_i| \leqslant 2\frac{\alpha^{1/\alpha}}{p^{1/p}}\sqrt{\frac{2\log n}{n}}) \asymp \frac{\alpha^{1/\alpha}}{p^{1/p}}\sqrt{\frac{2\log n}{n}},$$

which implies that

$$\frac{1}{n}\sum_{i=1}^{n} I(|X_i| \leqslant 2|v_i|) \asymp \frac{\alpha^{1/\alpha}}{p^{1/p}}\sqrt{\frac{\log n}{n}}.$$

As a result, we have that when $p > 1$,

$$\sum_{i:|X_i|\leqslant 2|v_i|} (|X_i + v_i|^p - |X_i|^p) \lesssim \frac{\alpha^{1/\alpha}}{p^{1/p}}\sqrt{n\log n} \cdot (\frac{\log n}{\sqrt{n}})^p = \frac{\alpha^{1/\alpha}}{p^{1/p}}n^{(1-p)/2}(\log n)^{p+1/2} = o(1).$$

Similarly, for Lemma 5.4, we can use the same idea to show

$$\sum_{|X_i|<2|v_i|} p\,\text{sgn}(X_i)|X_i|^{p-1}v_i = o_P(1).$$

In fact, by using the same derivation, we have

$$\sum_{|X_i|<2|v_i|} p\,\text{sgn}(X_i)|X_i|^{p-1}v_i \lesssim \frac{\alpha^{1/\alpha}}{p^{1/p}}\sqrt{n\log n} \cdot (\frac{\log n}{\sqrt{n}})^p = \frac{\alpha^{1/\alpha}}{p^{1/p}}n^{(1-p)/2}(\log n)^{p+1/2} = o(1).$$

## D.5  $p = 1$

We now study the case where $p = 1$. Since, for general $p$, we have

$$\tilde{\varphi}_n(x) = c_{p,\alpha}n^{1-\frac{\alpha}{p}}\|x\|_p^{\alpha},$$

where $c_{p,\alpha} = \alpha^{-1} \cdot p^{-\alpha+\frac{\alpha}{p}} \cdot \left(\frac{\Gamma(2-\frac{1}{p})}{\Gamma(\frac{1}{p})}\right)^{-\frac{\alpha}{2}}$.

Letting $p = 1$ we get

$$\tilde{\varphi}_n(x) = \frac{1}{\alpha}n^{1-\alpha}\|x\|_1^{\alpha}.$$

We first study the limit of $\|X\|_1$ when the density of $X$ is given by $\frac{1}{Z}e^{-\tilde{\varphi}_n(x)}$.

According to Theorem C.5, we have

$$X \overset{d}{=} t^{1/\alpha} \cdot r \cdot U,$$

where $t \sim \Gamma(\frac{n}{\alpha} + 1, 1)$, $U$ has uniform distribution over $\partial K_\varphi = \{x : \|x\|_1 = (\alpha n^{\alpha-1})^{1/\alpha}\}$, $r$ has density $nx^{n-1}$ over $[0, 1]$, and $U, r, t$ are independent.

As a result,

$$\|X\|_1 = |t^{1/\alpha}| \cdot |r| \cdot (\alpha n^{\alpha-1})^{1/\alpha} \sim (\frac{n}{\alpha})^{1/\alpha} \cdot 1 \cdot (\alpha n^{\alpha-1})^{1/\alpha} = n.$$

Now let us consider $\nabla\tilde{\varphi}_n(x)$, and we have

$$\nabla\tilde{\varphi}_n(X) = \frac{1}{\alpha}n^{1-\alpha} \cdot \alpha \cdot \|x\|_1^{\alpha-1} \cdot sgn(X) = sgn(X).$$

Since $X$ is symmetric, the above expression implies the **Condition D3**, that is, $\|\nabla\tilde{\varphi}_n(X)\|_2 \sim \sqrt{n}$ and therefore $v^\top\nabla\tilde{\varphi}_n(X) \to N(0,1)$.

To prove **Condition D1**, since $\frac{1}{2}vI_\varphi v = \frac{1}{2}$, it suffices to show when $\|v\|_2 = 1, \alpha = 1$,

$$\|X + v\|_1 - \|X\|_1 \to N(\frac{1}{2}, 1).$$

The case where $\alpha > 1$ can be reduced to this setting by using the following technique. Let us write

$$X \overset{d}{=} t^{1/\alpha} \cdot r \cdot U,$$

where $t \sim \Gamma(\frac{n}{\alpha} + 1, 1)$, $U$ has uniform distribution over $\partial K_\varphi = \{x : \|x\|_p = n^{1-1/\alpha}\}$, $r$ has density $nx^{n-1}$ over $[0,1]$, and $U, r, t$ are independent.

$$\tilde{X} \overset{d}{=} \tilde{t} \cdot r \cdot \tilde{U},$$

where $\tilde{t} \sim \Gamma(n+1, 1)$ and $\tilde{U} = n^{1/\alpha-1}U$.

Then

$$\sum_i(|X_i + v_i| - |X_i|) \sim t^{1/\alpha} \cdot r \cdot \sum_i(|U_i + (\frac{n}{\alpha})^{-1/\alpha}v_i| - |U_i|)$$

$$\sim t^{1/\alpha} \cdot r \cdot n^{1-1/\alpha}\sum_i(|\tilde{U}_i + n^{-1} \cdot \alpha^{1/\alpha}v_i| - |\tilde{U}_i|)$$

$$\sim (\frac{1}{\alpha})^{1/\alpha} \cdot \tilde{t}^{1/\alpha} \cdot r \cdot n^{1-1/\alpha}\sum_i(|\tilde{U}_i + n^{-1} \cdot \alpha^{1/\alpha}v_i| - |\tilde{U}_i|)$$

$$\sim (\frac{1}{\alpha})^{1/\alpha} \cdot n^{1-1/\alpha}\sum_i(|\tilde{X}_i + n^{1/\alpha-1} \cdot \alpha^{1/\alpha}v_i| - |\tilde{X}_i|),$$

which reduces to the $\alpha = 1$ setting up to some scaling.

Therefore, it suffices to show the asymptotic normality of $\|X + v\|_1 - \|X\|_1$ when $X$ has density $\propto e^{-\|X\|_1}$. We are going to use the Berry-Esseen theorem. Suppose we have $n$ independent random variables $X_1, \ldots, X_n$ with $\mathbb{E}X_i = \mu_i, \text{Var}X_i = \sigma_i^2, \mathbb{E}|X_i - \mu_i|^3 = \rho_i^3$. Consider the normalized random variable

$$S_n := \frac{\sum_{i=1}^n X_i - \mu_i}{\sqrt{\sum_{i=1}^n \sigma_i^2}}.$$

Denote its cdf by $F_n$. Then

**Theorem D.3** (Berry-Esseen). *There exists a universal constant $C > 0$ such that*

$$\sup_{x\in\mathbb{R}}|F_n(x) - \Phi(x)| \leqslant C \cdot \frac{\sum_{i=1}^n \rho_i^3}{\left(\sum_{i=1}^n \sigma_i^2\right)^{\frac{3}{2}}}.$$

In the following, we proceed to calculating these three moments for the random variable $X_1, \ldots, X_n \sim p$ with density $p(x) = \frac{1}{2}e^{-|x|}$ for $x \in \mathbb{R}$.

### D.5.1 Expectation

Without loss of generality we assume $v > 0$.

$$\mathbb{E}[|X+v|] - \mathbb{E}[|X|] = \frac{1}{2} \int_{-\infty}^{\infty} |x+v| e^{-|x|} \, dx - \frac{1}{2} \int_{-\infty}^{\infty} |x| e^{-|x|} \, dx$$

$$= \frac{1}{2} \left( \int_{-\infty}^{-v} |x+v| e^{-|x|} \, dx + \int_{-v}^{0} |x+v| e^{-|x|} \, dx + \int_{0}^{\infty} |x+v| e^{-|x|} \, dx - \int_{-\infty}^{0} |x| e^{-|x|} \, dx - \int_{0}^{\infty} |x| e^{-|x|} \, dx \right)$$

$$= \frac{1}{2} \left( \int_{-\infty}^{-v} -(x+v) e^{x} \, dx + \int_{-v}^{0} (x+v) e^{x} \, dx + \int_{0}^{\infty} (x+v) e^{-x} \, dx + \int_{-\infty}^{0} x e^{x} \, dx - \int_{0}^{\infty} x e^{-x} \, dx \right)$$

$$= \frac{1}{2} \left( \int_{v}^{\infty} x e^{-x} \, dx - \int_{0}^{v} x e^{-x} \, dx - \int_{0}^{\infty} x e^{-x} \, dx + v \left( -\int_{v}^{\infty} e^{-x} \, dx + \int_{0}^{v} e^{-x} \, dx + \int_{0}^{\infty} e^{-x} \, dx \right) \right)$$

$$= -\int_{0}^{v} x e^{-x} \, dx + v \int_{0}^{v} e^{-x} \, dx$$

$$= -(1 - e^{-v} - v e^{-v}) + v(1 - e^{-v})$$

$$= (v - 1 + e^{-v}) = \frac{1}{2} v^2 + o(v^2)$$

### D.5.2  Variance

$$2\mathbb{E}[|X+v| \cdot |X|] = \int_{-\infty}^{\infty} |x+v| \cdot |x| e^{-|x|} \, dx$$

$$= \int_{-\infty}^{-v} |x+v| \cdot |x| e^{-|x|} \, dx + \int_{-v}^{0} |x+v| \cdot |x| e^{-|x|} \, dx + \int_{0}^{\infty} |x+v| \cdot |x| e^{-|x|} \, dx$$

$$= \int_{-\infty}^{-v} (x+v) \cdot x e^{x} \, dx - \int_{-v}^{0} (x+v) x e^{x} \, dx + \int_{0}^{\infty} (x+v) \cdot x e^{-x} \, dx$$

$$= \int_{-\infty}^{-v} x^2 e^{x} \, dx - \int_{-v}^{0} x^2 e^{x} \, dx + \int_{0}^{\infty} x^2 e^{-x} \, dx + v \left( \int_{-\infty}^{-v} x e^{x} \, dx - \int_{-v}^{0} x e^{x} \, dx + \int_{0}^{\infty} x e^{-x} \, dx \right)$$

$$= 2 \int_{v}^{\infty} x^2 e^{-x} \, dx + 2v \cdot \int_{0}^{v} x e^{-x} \, dx$$

$$= 2v^2 e^{-v} + 4v e^{-v} + 4 e^{-v} + 2v \cdot (1 - v e^{-v} - e^{-v})$$

$$= 2v + 2v e^{-v} + 4 e^{-v} \sim 4 + o(v^2)$$

$$2\mathbb{E}[|X+v|^2] = \int_{-\infty}^{\infty} (x+v)^2 e^{-|x|} \, dx$$

$$= \int_{-\infty}^{\infty} x^2 e^{-|x|} \, dx + v^2 \int_{-\infty}^{\infty} e^{-|x|} \, dx$$

$$= 2 \int_{0}^{\infty} x^2 e^{-x} \, dx + 2v^2 \int_{-\infty}^{\infty} e^{-x} \, dx$$

$$= 4 + 2v^2$$

$$2\mathbb{E}[|X|^2] = \int_{-\infty}^{\infty} |x|^2 e^{-|x|} \, dx$$

$$= 2 \int_{0}^{\infty} x^2 e^{-x} \, dx$$

$$= 4$$

$$2\mathbb{E}[(|X+v|-|X|)^2] = \mathbb{E}[|X+v|^2] + \mathbb{E}[|X|^2] - 2\mathbb{E}[|X+v|\cdot|X|]$$
$$= \frac{1}{2}(8 + 2v^2 - 2(4 + o(v^2)))$$
$$= v^2 + o(v^2)$$

### D.5.3   Third moment

$$\mathbb{E}[|X|^3] = \int_{-\infty}^{\infty} |x|^3 e^{-|x|}\, dx$$
$$= 2\int_0^{\infty} x^3 e^{-x}\, dx$$
$$= 12$$

$$2\mathbb{E}[|X+v|^3] = \int_{-\infty}^{\infty} |x+v|^3 e^{-|x|}\, dx$$
$$= \int_{-\infty}^{-v} |x+v|^3 e^{-|x|}\, dx + \int_{-v}^{0} |x+v|^3 e^{-|x|}\, dx + \int_0^{\infty} |x+v|^3 e^{-|x|}\, dx$$
$$= \int_{-\infty}^{-v} -(x+v)^3 e^x\, dx + \int_{-v}^{0} (x+v)^3 e^x\, dx + \int_0^{\infty} (x+v)^3 e^{-x}\, dx$$
$$= 6e^{-v} + (-6 + 6e^{-v} + 6v - 3v^2 + v^3) + (v^3 + 3v^2 + 6v + 6)$$
$$= 12e^{-v} + 12v + 2v^3$$

$$2\mathbb{E}[|X+v|^2 \cdot |X|] = \int_{-\infty}^{\infty} (x+v)^2 \cdot |x| e^{-|x|}\, dx$$
$$= -\int_{-\infty}^{0} (x+v)^2 x e^x\, dx + \int_0^{\infty} (x+v)^2 \cdot x e^{-x}\, dx$$
$$= (v^2 - 4v + 6) + (v^2 + 4v + 6)$$
$$= 2v^2 + 12$$

$$2\mathbb{E}[|X+v|\cdot|X|^2] = \int_{-\infty}^{\infty} |x+v| \cdot x^2 e^{-|x|}\, dx$$
$$= -\int_{-\infty}^{-v} (x+v) \cdot x^2 e^x\, dx + \int_{-v}^{0} (x+v)x^2 e^x\, dx + \int_0^{\infty} (x+v)x^2 e^{-x}\, dx$$
$$= e^{-v}(v^2 + 4v + 6) + [2v - 6 + e^{-v}(v^2 + 4v + 6)] + 2(v + 3)$$
$$= 4v + 2e^{-v}(v^2 + 4v + 6)$$

$$\mathbb{E}[(|X+v|-|X|)^3] = \frac{1}{2}(-12 + 12e^{-v} + 12v + 2v^3 - 3(2v^2 + 12) + 3(4v + 2e^{-v}(v^2 + 4v + 6)))$$
$$= \frac{1}{2}(-48 + 48e^{-v} + 24v + 2v^3 - 6v^2 + 6v^2 e^{-v} + 24ve^{-v})$$
$$= \frac{1}{2}(48(-v + v^2/2 + o(v^2)) + 24v + 2v^3 - 6v^2 + 6v^2(1 - v + o(v)) + 24v(1 - v + o(v)))$$
$$= o(v^2)$$

Combining the pieces, we get

$$\sum_{i=1}^{n} \mathbb{E}[|X_i + v_i| - |X_i|] = \frac{1}{2} \sum_{i=1}^{n} v_i^2 + o(1) = \frac{1}{2} + o(2)$$

$$\sum_{i=1}^{n} Var[|X_i + v_i| - |X_i|] = \sum_{i=1}^{n} v_i^2 + o(1) = 1 + o(1)$$

$$\sum_{i=1}^{n} \mathbb{E}[(|X_i + v_i| - |X_i| - \mathbb{E}[|X_i + v_i| - |X_i|])^3] = o(\sum_{i=1}^{n} v_i^2) = o(1).$$

Therefore, by using Theorem D.3, we get the desired result.