# OpenReview forum: "A Central Limit Theorem for Differentially Private Query Answering"
_NeurIPS.cc/2021/Conference — NeurIPS 2021 Spotlight_

### Official Review · Reviewer_Dzvr · 2021-06-30

**Rating:** 6
**Confidence:** 3

**Summary:**

Within the framework of f-DP, the paper establishes a CLT for high-dimensional additive noise mechanims, demonstrating that as the dimension increases, adding independent noise to the coordinates has a privacy loss similar to Gaussian noise. This result addresses the question of what noise mechanism is best applied, indicating that Gaussian noise offers the best privacy-utility tradeoff. The privacy-utility tradeoff is communicates through an "uncertainty principle", which shows that the product of the privacy parameter and a measure of utility are bounded, indicating that both quantities cannot be simultaneously small. They show that Gaussian noise makes this uncertainty inequality tight.


**Ethical Concerns:**

No issues.

**Limitations And Societal Impact:**

No issues.

**Main Review:**


Originality:
The notion of a CLT in terms of the privacy guarantee when many mechanisms are composed was previously established in [DRS21]. The setting considered in this paper is similar to that of [DRS21], but slightly different. For composition of mechanisms, a privacy mechanism is applied sequentially to each statistic of interest, and quantities such as sensitivity are usually evaluated for each statistic separately. In the high-dimensional setting of this paper the L_2 sensitivity of the statistic vector is assumed to be bounded, and a noise vector is simultaneously added to the statistic vector. While the setting does not seem to be contained within the setup of [DRS21], it is still fairly similar. As such, the result that the tradeoff function converges to that of a Gaussian is not very surprising.

However, besides the work [DRS21], this paper is one of few to use classical statistics techniques (such as CLT, FIsher Information) to analyze privacy problems. As such the techniques of the paper are fairly unique among DP works.


Quality:
The claims are supported with mostly theoretical results, demonstrating the CLT phenomenon and a unique privacy-utility tradeoff. Some simulations are also given to illustrate the convergence of the ROC curve to that of a Gaussian for the statistic dimension n=30.



Significance:
Determining privacy mechanisms that optimize the privacy/utility tradeoff are a common and important problem in DP. This paper approaches this problem from a unique perspective, looking at the privacy cost of i.i.d. additive noise as the dimension of the statistic increases. They argue from their framework that if i.i.d. additive noise is used for privacy, then Guassian noise is the ideal distribution as it makes the privacy/utility tradeoff tight.

An unfortunate problem of CLT arguments in terms of privacy cost is that the results often cannot be used for rigorous privacy guarantees. The authors note this fact, and state that the purpose of the CLTs is to highlight an interesting phenomenon, which shows that Gaussian noise offers a sharp privacy-utility tradeoff. While this makes sense, it does limit the appeal of the CLT results, since they cannot be used to argue the privacy cost of additive noise.

A limitation of the scope of the paper is that the additive mechanisms considered are assumed to be i.i.d. for each entry of the noise vector. This removes the possibility of adding correlated noise. I suspect that if correlated noise were an option, then it would be possible to construct additive mechanisms whose tradeoff functions do not approach G_mu. For example, consider the K-norm mechanisms:

Hardt, Moritz, and Kunal Talwar. "On the geometry of differential privacy." Proceedings of the forty-second ACM symposium on Theory of computing. 2010.

Awan, Jordan, and Aleksandra Slavković. "Structure and sensitivity in differential privacy: Comparing k-norm mechanisms." Journal of the American Statistical Association (2020): 1-20.

Clarity:
The presentation of the paper is somewhat difficult to read. Some of the mathematical results and conditions are difficult to parse (such as (D1) and (D2), as well as Theorem 3.1 and Corollary 3.2. The exposition does help in certain cases, such as in Remarks 1 & 2.

However, in other places the exposition is also difficult to understand. For example Lines 188-191, it is not clear what is meant at this point in the text by "the normalized version". Lines 306-314 is also a confusing exposition about uncertainty principles, which seems to be tangent to the main discussion (It might be helpful to put this discussion into a remark, or to postpone it to discussion of Section 5).

Another point that is not made very clear in the exposition is that the additive mechanisms considered in the paper are only those that add i.i.d. noise to the entries of the statistic vector. This rules out correlated noise such as discussed under "Significance".

There are two missing equation numbers on page 8.

The parentheses in (D1) are unbalanced.

Conclusions:
The paper builds off of [DRS21] to develop similar CLT results for additive mechanisms in high dimensions. These results are interesting in that they show that as the dimension goes to infinity, adding i.i.d. noise often results in a privacy cost similar to Gaussian noise. A related result shows that the privacy-utility tradeoff is optimized by using Gaussian noise. However, these results are very similar to those of [DRS21] (in fact, if L_infinity sensitivity were used in place of L_2, then the framework would fit within that of [DRS21]). Due to this, the conclusions are not very surprising.

Furthermore, these results are mostly of theoretical interest, and do not result in many practical results. One takeaway is that if i.i.d. noise is to be added, then the Gaussian mechanism is optimal in some sense. However, this is already a commonly used mechanism, so the results do not result in any new DP mechanisms. Furthermore, as the authors mention in Line 92-99, the CLT cannot be used to rigorously quantify the privacy loss when using additive noise (from non-Gaussian distributions).

Another issue with the paper is the presentation. While the main ideas came across, the paper was difficult to read, with some sections very difficult to parse. The paper would benefit from additional revision.

In its present form, the paper is marginally below the acceptance threshold (5).

=======
Update: Thanks to the authors for clarifying that the paper allows for correlated noise, and is significantly different from the results of DRS21. I update my score from 5 to 6. I encourage the authors to revise the exposition to improve clarity, if the paper is accepted.

**Time Spent Reviewing:**

2.5

---

> ### Author Response · Authors · 2021-08-10
> **Author Response**
>
> Thank you for the detailed review and complimentary words.
>
> We would like to point out a serious misunderstanding: our results allow *correlated* noise. In fact, in line 67, immediately after stating Theorem 1.1 (our CLT), we point out that the CLT holds when the noise has density proportional to $\mathrm{e}^{-\\|x\\|_p^\alpha}$ for any $p,\alpha\geqslant 1$, which is generically non-iid unless $p=\alpha$. This is what convinced us that the CLT phenomenon should occur for general log-concave noises (Conjecture 3.4), including the K-norm mechanism for all K.
>
> However, we agree that this point is worth of a more detailed discussion. Indeed, if our result only handled the iid case, it would be an immediate consequence of the compositional CLT in [DRS21] and hence far less interesting. For general correlated noise (e.g. the above with $p\neq\alpha$), a bit of effort clearly suggests that the compositional CLT in [DRS21] cannot be applied. We have appended a short paragraph of explanation  at the end of the response and plan to include it as an independent section of the appendix.
>
> ---
>
> Regarding the rigorousness of the privacy guarantee offered by the CLT approach, we treat it as one of the following two desiderata in the practice of DP:
> 1. "small constant"
> 2. "security level of rigorousness"
>
> By "security level of rigorousness" we mean a mathematically rigorous finite bound, excluding the use of any limit such as CLT.
> By "small constant" we refer to the "constant" factor in front of the privacy parameters of various mechanisms. This factor is getting more and more attention as it can have crucial impact on the statistical utility of the mechanisms and hence the vitality of the entire field.
> After all, data is often a scarce and valuable resource, and this situation is unfortunately more common in DP-relevant scenarios than others.
>
> Unfortunately, there are very few (if any) examples where these two desiderata can be achieved perfectly at the same time. Most of the existing work emphasize the second, with great success in providing theoretical understandings, while how to achieve the first is relatively poorly understood. We treat our work as a novel attempt towards the first desiderata, and look forward to more works in this direction and ultimately the prosperous of the field in the long run.
>
> Hopefully the above comment also resolves your concern about the paper not resulting in many practical results. We think that the "small constant" desiderata might require "rethinking" the privacy-accuracy trade-off, and this is our attempt.
>
>
> ---
>
> Regarding the writing, your feedback on "the normalized version" is very valuable. We will change the wording. Regarding the classical uncertainty principle (line 306-314), we were concerned that the comparison might be confusing for readers who are only aware of the specific Heisenberg uncertainty principle $\Delta x \cdot \Delta p\geqslant \\frac{\hbar}{2}$. The analogy is far from obvious in this form and only manifests itself in the Fourier form. Putting into a remark seems a good idea to avoid distraction.
>
>
>
> ---
>
> **Appendix**
>
> Here, we explain why the compositional CLT in [DRS21] does not apply in our setting. Let us consider a two-dimensional noise-addition mechanism $M(D)=f(D)+{X \choose Y}={f_1(D)+X \choose f_2(D)+Y}$ where $X\choose Y$ has joint density $p(x,y)$. We denote the conditional density as $p(y|x)$. In order to treat $M$ as a two-step composition $M=(M_1,M_2)$, since $X$ and $Y$ are correlated, it is necessary that the second step $M_2$ takes the first output $x=M_1(D)$ as input. Specifically, $M_1(D)=f_1(D)+X$ and $M_2(D,x)=f_2(D)+Z_x$ for some noise $Z_x$. In order to have $M=(M_1,M_2)$, we need $Z_x\stackrel{d}{=}Y$ when $x=M_1(D)=f_1(D)+X$. Therefore, $Z_x$ has density $p(y|x-f_1(D))$ and $M_2(D,x)$ has density $p(y-f_2(D)|x-f_1(D))$.
>
> All composition theorems (including the one in [DRS21]) ask that $M_2(\cdot,x)$ is DP in the worst case of $x$. So we need to compare the densities of $M_2(D,x)$ and $M_2(D',x)$ for arbitrary $x\in\mathbb{R}$. However, the two densities $p(y-f_2(D)|x-f_1(D))$ and $p(y-f_2(D')|x-f_1(D'))$ can be arbitrarily far away in the worst case of $x$, which renders the composition an invalid approach. Even if it worked in some special cases that limits the amount of correlation, it would only be an upper bound and cannot extract Fisher information as the correct limit and hence cannot explain our numerical finding in Figure 1 of the paper. Therefore, our results differs from [DRS21] in a very non-trivial way.

---

> > ### Comment · Reviewer_Dzvr · 2021-08-19
> > **Misunderstanding**
> >
> > Thank you for the clarification. I know see that I misunderstood that you are allowing correlated noise in your result, which I agree makes it significantly different from the results of DRS21.

---

### Official Review · Reviewer_od6Q · 2021-07-20

**Rating:** 7
**Confidence:** 3

**Summary:**

This paper considers Differential Privacy on multidimensional query responses and in particular considers the hypothesis testing interpretation of Differential Privacy.  The authors rely heavily on results derived in the cited paper DRS21 to derive bounds on highdimensional queries

**Limitations And Societal Impact:**

Yes

**Main Review:**

The authors place their work in the framework of f-DP and the Gaussian Differential Privacy defintion.  They claim to show that a new interpretation of noise addition and the hypothesis testing approach when the query dimensions are very high.  In particular they show that a large class of noise addition mechanisms fall in the Gaussian mechanism framework when the dimension is large. The novelty here is not clear.  The authors should show why the previous resutls (from DRS21 for example) do not extend to this case, since the framework from DRS21 is quite large.

While the result on high dimensions is nice, the paper leaves a gap in practical understanding.  Some experiments that show how the theoretical results manifest would be needed.

The authors make some comments on similarity to Uncertainty Principles, but this section is unclear - why is this important and what's the significance?  This is a different approach to the privacy-accuracy tradeoff, and how does the interpretation of the Uncertainly Principle help here?


**Time Spent Reviewing:**

4

---

> ### Author Response · Authors · 2021-08-10
> **Author response**
>
> Thank you for your time and for the overall positive review.
>
> Regarding your comments on the novelty, we would like to point out that this result **does not** follow from [DRS21], which only handles composition. In fact, it significantly extends the CLT there. The main reason is that we allow **correlated** noise. At this point it might be helpful to highlight the fact that despite the same name, Gaussian limit in high dimensional spaces under correlation is a much more far-reaching phenomenon than textbook CLT for iid random variables. The ultimate achievement [Klartag 2007], which motivates our work and our conjecture, has been a remarkable success in functional analysis and published in Inventiones Mathematicae.
>
> However, we agree that this point is worth of a more detailed discussion. Indeed, if our result only handled the iid case, it would be an immediate consequence of the compositional CLT in [DRS21] and hence far less interesting. For general correlated noise (e.g. the above with $p\neq\alpha$), a bit of effort clearly suggests that the compositional CLT in [DRS21] cannot be applied. We have appended a short paragraph of explanation at the end of the response and plan to include it as an independent section of the appendix.
>
> **Reference**
>
> Klartag, Bo'az. "A central limit theorem for convex sets." Inventiones mathematicae 168.1 (2007): 91-131.
>
> ---
>
> Thanks for the comment on experiments. This indeed relatively less emphasized in the main text. In the appendix we demonstrated the elements of the simulation we used to observe the phenomenon and why sample size of 10,000 is enough. We also attached a Python notebook showing that a dimension of 30 is enough, and it also left room for future investigation. We are planning to incorporate the comparison of more noises in terms of both privacy and accuray.
>
> ---
>
> The importance of uncertainty principle lies in that it provides a simple and constant-sharp lower bound that characterizes the privacy-accuracy trade-off, which is the major goal of this paper. We draw connection to "uncertainty principle" because we think the Heisenberg uncertainty principle is well-known to be simple and constant-sharp. It might as well be termed as a "Cramer--Rao" type of lower bound. We apologize for the confusion and would be happy to modify the presentation.
>
> ---
>
> **Appendix**
>
> Here, we explain why the compositional CLT in [DRS21] does not apply in our setting. Let us consider a two-dimensional noise-addition mechanism $M(D)=f(D)+{X \choose Y}={f_1(D)+X \choose f_2(D)+Y}$ where $X\choose Y$ has joint density $p(x,y)$. We denote the conditional density as $p(y|x)$. In order to treat $M$ as a two-step composition $M=(M_1,M_2)$, since $X$ and $Y$ are correlated, it is necessary that the second step $M_2$ takes the first output $x=M_1(D)$ as input. Specifically, $M_1(D)=f_1(D)+X$ and $M_2(D,x)=f_2(D)+Z_x$ for some noise $Z_x$. In order to have $M=(M_1,M_2)$, we need $Z_x\stackrel{d}{=}Y$ when $x=M_1(D)=f_1(D)+X$. Therefore, $Z_x$ has density $p(y|x-f_1(D))$ and $M_2(D,x)$ has density $p(y-f_2(D)|x-f_1(D))$.
>
> All composition theorems (including the one in [DRS21]) ask that $M_2(\cdot,x)$ is DP in the worst case of $x$. So we need to compare the densities of $M_2(D,x)$ and $M_2(D',x)$ for arbitrary $x\in\mathbb{R}$. However, the two densities $p(y-f_2(D)|x-f_1(D))$ and $p(y-f_2(D')|x-f_1(D'))$ can be arbitrarily far away in the worst case of $x$, which renders the composition an invalid approach. Even if it worked in some special cases that limits the amount of correlation, it would only be an upper bound and cannot extract Fisher information as the correct limit and hence cannot explain our numerical finding in Figure 1 of the paper. Therefore, our results differs from [DRS21] in a very non-trivial way.

---

> > ### Comment · Reviewer_od6Q · 2021-09-10
> > **Clear**
> >
> > Thank you for the clarification.

---

### Official Review · Reviewer_S8w3 · 2021-07-21

**Rating:** 8
**Confidence:** 4

**Summary:**

This paper studies the privacy-accuracy trade-off of answering a single n-dimensional query with ell-2 sensitivity 1, under differential privacy. This problem has been studied and the optimal noise distribution the Gaussian, and its upper and lower error bounds match up to constant factors. This paper however studies optimality in the limit n->infty and shows that the Gaussian mechanism is optimal even with respect to constant factors.

A central quantity to the approach and intuition in this work is the ROC function: using a known hypothesis testing view of DP, the ROC function T[M(D),M(D′)] measures the optimal error distinguishing M(D) and M(D’) for two neighboring datasets D, D’ and we say that M is (epsilon, delta)-DP if this function is above a fixed function f_(epsilon, delta) pointwise. Let G be the ROC function of the Gaussian mechanism (omitting privacy parameters).

Starting from the 1-dimensional case, the authors demonstrate that the optimal curve f_(epsilon, delta) and G are far whereas for example the ROC function of the truncated Laplace mechanism is closer to the optimal tradeoff, which they use to stress that the Gaussian mechanism is wasteful in 1-dimension and ask whether this is also the case for n->infty.

The main contribution of the paper is a central limit type of theorem that demonstrates that this is not the case: for all distributions of noise that satisfy a certain set of conditions, as n->infty, inf_{D,D’} T[M_n(D),M_n(D’)] -> G pointwise. That is, the sequence of mechanisms with the specified noise distributions approach the tradeoff of the Gaussian mechanism. This holds for almost all neighboring pairs of datasets, with the probability of the event that it does not happen going to 0, as n->infty. These conditions are satisfied for example by noise distributions of the form exp(-||x||^a_p) for any ell-p norm and power a. For these and more generally noise distributions of the form exp(-φ) where φ has finite second moments and Fisher information matrix, the authors also prove that the privacy parameter times the ell-2 error of any mechanism is lower bounded by the dimension n, with the inequality being tight when we choose the n-dimensional Gaussian as the noise distribution.

**Limitations And Societal Impact:**

I think that the potential negative societal impact section is not very informative as is (it actually doesn't include any potential negative societal implications) and might suggest removing it altogether as it does not seem applicable to this work.

Limitations are discusses.

**Main Review:**

*Strengths:*
- The paper takes a novel creative perspective to the privacy-accuracy trade-off and the study of the optimal noise addition mechanism for high-dimensions.
- It is hard to tell which parts of the proofs might be useful for future research, but most of the proofs are not trivial and more importantly there are a few ideas and quantities/notions that seem potentially useful (such as the “likelihood projection”).

*Weaknesses:*
- The results would be described as being of “purely” theoretical nature as they study the asymptotic behavior of the ROC curve of DP mechanisms. It is not straightforward to derive insights as to which noise distributions are constant-sharp optimal for finite -but high- dimensions and we already knew that the Gaussian mechanism is the right choice (up to constants). But the problem itself is important and the paper includes several insights (beyond the main ones described above), so I would not consider this to be an argument for rejection.


*Detailed comments:*
- Although there are explanations in the paper, the regularity conditions are hard to interpret. Are there more examples of phi (leading to known noise distributions that have been used in the literature) that satisfy these conditions?
- How does this work extend ENU? The latter refers to factorization mechanisms and not more generally noise addition mechanisms but the result is stated as a characterization of the the sample complexity rather than an asymptotic result. It might be beneficial to add more discussion on this in the related work section.
- I found that the presentation of the main paper was successful in conveying the high level picture and rationale behind the contributions. The proofs however were not as easy to read, as not all proofs and/or lemma statements are close to where they are used. I would suggest adding subsections to group results that are combined in previously stated proofs or use Claims/Propositions within the proofs of Lemmas for results that are needed. The current write-up seems to state both the “small” and the “big” lemmas one after the other.
-Abstract: perhaps add “for some alpha, p” as one would expect to have seen the variables before the relevant statement
- line 37: Put it differently -> Put differently
- line 96: sharp in constant -> sharp with respect to constants (?)
- line 119: privcacy -> privacy
- line 146: explain the embarrassing failure of  Gaussian mechanism: perhaps this is a bit excessive…?
- line 172: N(f(D),…) -> N(f(D’),…)
- line 188: suggests, the -> suggests that the
- line 305,318: broken reference
- line 464: I think the left parenthesis should be before 1+…
- line 527: I did not understand why we also need the opposite of the bad sets of vectors -Theta_n and -Omega_n in the event E_n.
- There is a mismatch in almost all the appendix, where references are for section C,D,E but really refer to sections B,C,D, respectively.
- line 708: I assumed o_P is as defined in the main body but P has a different font in the appendix.
- line 720: equation should be split


==========


Thanks to the authors for their response and for providing examples. I still support acceptance.

**Time Spent Reviewing:**

7

---

> ### Author Response · Authors · 2021-08-10
> **Author response**
>
> Thank you for the very encouraging and positive comments!
>
> Regarding the regularity conditions, there are examples in the literature. For example, the K-norm mechanism [Hardt and Talwar, 2010] is a special case if we take $\varphi$ to be the $K$-norm. The recent progress in answering queries with small $\ell_\infty$ error [Ganesh and Zhao 2020] uses $\varphi(x) = \\|x\\|_p^p$ and they call it "generalized Gaussian".
>
> ---
>
> Results in ENU have two major differences:
> 1. As we pointed out in the related work paragraph, they consider a refined version of sensitivity that depends on the specific "workload", which, in particular, is only defined for linear queries.
> 2. To the best of our knowledge, the optimality of Gaussian mechanism is up to a $\log\delta^{-1}$ factor. For comparison, the factorization mechanism is rate-optimal, but is not optimal in terms of constant. Such a constant factor (not to mention log factors) is what we try to understand and eliminate in this work.
>
> Indeed, because of the strong convex geometry flavor in both works, the connection should be discussed in more detail. Thanks for the suggestion.
>
> ---
>
> Thank you for your suggestion for organizing the proofs. We will re-organize the proof for clarity.
>
> ---
>
> We appreciate your patience in picking up our typos. We will correct them if given the chance.
>
> Regarding the minus $\Theta_n$, recall that in the equation between line 527 and 528, we need the inequality for both $v$ and $-v$, while $\Theta_n$ is the bad set for $v$. That is why we need $v$ to avoid both $\Theta_n$ and $-\Theta_n$.

---

### Official Review · Reviewer_JiKW · 2021-07-22

**Rating:** 7
**Confidence:** 1

**Summary:**

The paper offers a central limit theory perspective to study two commonly used algorithms in the DP community Gaussian and Laplace for query answering. They consider two scenarios: when the dimension is one (that is there is only one query to answer) and very high dimension (when the answer vector is very hight dimensional). As I understand this, the theory tries to explain the optimality of the Gaussian mechanism in the high dimensional setting even thought it seems to be suboptimal in the one dimensional case in terms of l-2 error.

**Limitations And Societal Impact:**

yes

**Main Review:**


The show that a large number of noise addition mechanisms behave like the Gaussian mechanism and they prove that the Gaussian mechanism achieves the optimal sharp-constant private-accuracy trade-off. I find the analysis and the insights in this paper very interesting although it did not understand all the claims.


**Time Spent Reviewing:**

1

---

> ### Author Response · Authors · 2021-08-10
> **Author response**
>
> Thank you for the positive review and finding our paper interesting.  We will incorporate your comments and polish the paper to make the message more accessible.

---

### Decision · Program_Chairs · 2021-09-27

**Decision:**

Accept (Spotlight)

**Comment:**

The authors theoretically analyze optimal privacy-utility tradeoffs when answering an n-dimensional query under differential privacy as n grows large, and provide new evidence pointing to the optimality of the Gaussian mechanism. Four expert reviewers voted accept, with scores 6, 7, 7, 8, and made overall very positive comments about the paper. Two reviewers described raised questions about the relationship between this paper and [DRS21]. The authors clarified that this paper considers correlated noise, while [DRS21] considers only independent noise, which satisfied the concerns of one reviewer, who raised their score. This point is probably worth clarifying in the final revision.